# Path Following of an Underwater Snake-like Robot Exposed to Ocean Currents and Locomotion Efficiency Optimization Based on Multi-Strategy Improved Sparrow Search Algorithm

Jing Liu [ID], Haitao Zhu *, Yan Chen and Han Bao [ID]

College of Shipbuilding Engineering, Harbin Engineering University, No. 145 Nantong Street,
Harbin 150001, China; 120010008@hrbeu.edu.cn (J.L.); s321517361@163.com (Y.C.); baohan@hrbeu.edu.cn (H.B.)
* Correspondence: zhuhaitao@hrbeu.edu.cn

**Abstract:** In this paper, we propose a cubic spline interpolation method to generate a desired curve path and present an integral line of sight (ILOS) method and a control strategy for course tracking control based on nonsingular terminal sliding mode to enable an underwater snake-like robot (USR) to move towards and follow the path generated by the parametric cubic-spline interpolation (PCSI) path-planning method, while considering the disturbances caused by ocean currents. The efficiency of robot locomotion is an important evaluation criterion for robot design. Thus, we introduce a multi-strategy improved sparrow search algorithm (MISSA) to dynamically choose gait parameters that significantly enhance the efficiency of robot movement. We conduct simulations to demonstrate that the proposed controller enables the USR, subjected to ocean currents, to accurately converge towards and follow the target path. Our results also reveal that MISSA effectively enhances the locomotion efficiency of the robot.

**Keywords:** path following; underwater snake-like robot; multi-strategy improved sparrow search algorithm; locomotion efficiency

## 1. Introduction

Over the past few decades, the development of underwater vehicles has gained importance due to their growing role in exploring and exploiting ocean resources. However, conventional framed robots face difficulties in narrow underwater working areas. To overcome these challenges, engineers and scientists have sought new solutions, including the design of more robust and slender robots. Natural species that have adapted to confined subsea environments have provided inspiration for the development of underwater biomimetic robots, such as fish-like and snake-like robots, that can operate effectively in these environmental constraints by leveraging their unique physical features. There has been a growing body of academic research on these robots over the years, although the focus in underwater snake-like robots has primarily been on two-dimensional planes. Nevertheless, some prototypes of snake-like robots have been developed [1–3]. As science and technology continue to advance, the deployment of underwater snake-like robots in various work environments, such as seabed oil and gas industry exploration and the inspection and maintenance of offshore international structures, is expected to increase. In order to enable these robots to operate effectively in unknown environments, it is necessary to develop more reliable control techniques and efficiency optimization methods.

The mechanical structure of an underwater snake-like robot is composed of a certain number of joints, which makes it more complex than other robots, with a more complex dynamics model and highly nonlinear characteristics. The serpenoid curve-based approach, originally proposed by Hirose [4], is widely used due to its ability to simplify the implementation and parameter tuning process. Many researchers have designed controllers for snake robots based on it [5]. The mathematical modeling of these robots has been

extensively studied in previous articles [6–9]. The D-H parametric method was proposed by Professors J. Denavit and R. S. Hertenberg in 1955 and is widely used in the spatial relations of robot linkages [10]. Therefore, it can also be applied to the kinematic modeling of snake robots. Xiaoqing Zuo et al. [11] used the D-H parametric method to model the kinematics of snake robots by studying the parameters of snake curves. Prof. Kelasidi [7] constructed a kinematic model of an underwater snake-like robot by using the geometric relations of adjacent connecting rods, analyzed the hydrodynamics of the connecting rods by Morrison's formula, and applied the Newton–Euler method to model the dynamics of the underwater snake-like robot. Due to the challenges associated with applying a complex model in controller design, a simplified model, converting the angle into a relative distance between links, is proposed in [12]. The authors demonstrated that the control of the robot's motion exhibits similar characteristics between a complex model and a simplified one. The movement of the underwater snake-like robot (USR) was examined based on this model, and the steady-state averaged velocity dynamics were presented in [13]. The construction of the D-H parametric method is inconsistent depending on the chosen coordinate system. Moreover, when the two rotation axes are nearly parallel, a pathological matrix will be generated by the D-H method, which makes it difficult to guarantee the accuracy of the control. The kinematic model of the snake robot constructed by Prof. Kelasidi using geometric relations will not be affected by this, and the process of the kinetic model built by the Newton–Euler method is relatively simple and easy to understand compared to the Lagrangian method, so Prof. Kelasidi's mathematical modeling of the snake robot is used as the mathematical model of the snake robot in this paper.

After formulating the simplified mathematical model for the USR, the design of an appropriate control method becomes essential. PID control is one of the most common control methods, and it is no exception when applied to underwater snake-like robots [7]. Researchers have demonstrated that controlling underwater snake-like robots using the same controls as ground robots is effective [14]. In addition, the sliding mode variable structure control (SMVSC) algorithm has emerged as a promising technique due to its simplicity, robustness, and reliability. Thus, SMVSC can be applied to the motion control of USR. To mitigate the impact of strong time-varying internal and external disturbances and to ensure accurate whole-body trajectory tracking, a control strategy based on SMVSC and a linear matrix inequality nonlinear disturbance observer was proposed in [15]. Additionally, to improve tracking accuracy and achieve finite-time convergence of tracking errors, terminal sliding mode control [16] and fast terminal sliding mode control [17] have been proposed. However, linear controllers such as PID control are not suitable for non-linear control systems such as USR, where approximating the non-linearity with linearity reduces accuracy. The application of terminal sliding mode control can lead to singularities in certain areas, resulting in infinite control inputs. In contrast, nonsingular terminal sliding mode control directly solves the existing terminal sliding mode control singularity problem from the sliding mode design by purposefully changing the switching function to achieve global nonsingular control of the system; at the same time, it inherits the finite-time convergence characteristics of the terminal sliding mode, which enables the control system to converge to the desired trajectory in finite time with high steady-state accuracy compared to the traditional linear sliding mode control. It is particularly suitable for non-linear and high-precision control. Therefore, a nonsingular terminal sliding mode control method is used in this paper to implement USR motion control.

With regard to the control of underwater snake-like robots, various types of controllers have been proposed in the literature, including those that (1) achieve forward and turning locomotion [18]; (2) not only achieving forward and turning locomotion, but also follow a desired path, including a straight line or a curved one [18–20]; and (3) enable straight-line path tracking in the presence of ocean currents [14]. In path-following control research for underwater snake-like robots, one approach to controlling the robot's heading is through the use of a line-of-sight (LOS) law [21]. In addition, the use of integral LOS has been mentioned in [7] to obtain the reference heading angle, and a Poincaré map has been

utilized to analyze stability in previous studies [19,22]. To enhance the path-following control of underwater snake-like robots, A. M. Kohl et al. proposed a controller based on a control-oriented model and provided formal stability proof for the closed-loop system [14]. However, the majority of the existing literature only concentrates on the performance of path-following controllers on straight paths. In practice, a more complex guidance strategy, such as waypoint guidance (WPG), is commonly used for marine vehicles. For instance, WPG was introduced for land-based snake-like robots in [23]. Moreover, WPG has been applied to underwater bio-mimetic robots [24], and previous research has utilized WPG to enable a USR to avoid obstacles along the path [25]. In order to generate a smooth path for the snake robot, various interpolation methods have been proposed. For instance, Dubin's path method was proposed in [26] to create a straight path between two waypoints. This method involves turning the robot before it reaches the waypoint, rather than having it pass through directly. Additionally, cubic Hermite interpolation (CHI) and parametric cubic-spline interpolation have been used to create smooth paths [20,27], which are well-suited for snake robot path planning. The parametric cubic-spline interpolation is to divide the original long sequence into several segments to construct multiple cubic functions, so that the segments have the property of second-order derivative continuity at the articulation, and it is because of this property that the third spline interpolation is adopted as the method of path generation for the underwater snake-like robot in this paper.

Living snakes propel themselves forward by twisting their bodies, while snake-like robots achieve forward motion by swinging each of their links. The selection of the gait parameters for the serpenoid curve, as proposed by Hirose, is therefore crucial in affecting the locomotion of the robot. In a study investigating the relationship between the robot's forward speed and locomotion energy, three different locomotion gaits were examined [28]. Using the central pattern generator (CPG) model, Cao et al. [29] applied a cuckoo search algorithm to optimize the robot's parameters under varying environmental conditions, using a displacement-dissipation ratio to evaluate energy efficiency. However, the limitation of their approach is that it is only suitable for adjusting dynamic parameters in a defined environment. To investigate the relationship between forward speed and average power under various parameter combinations, a particle swarm optimization algorithm was applied, but this study only considered the case of a straight path [30].

The development of bionic intelligent algorithms has provided new solutions for optimization problems, which can achieve real-time computation due to their fast iteration speed. D. K. Dewangan has optimized the neural network using a bionic intelligence algorithm, which has greatly improved the accuracy of the output of the neural network [31,32]. An improved pigeon-inspired algorithm has been proposed to optimize robot system parameters in real time, aiming to overcome the challenges of slow convergence speed and susceptibility to local optimum [33]. Recently, Xue Jiankai proposed a new bionic intelligent algorithm, the sparrow search algorithm (SSA), inspired by the predation behavior of sparrow populations in nature [34]. The SSA has several advantages, such as fewer control parameters, high realization efficiency, and fast convergence speed. To further enhance the local search ability of the SSA algorithm, researchers have introduced tent chaotic sequence and Gaussian mutation into the sparrow population and perturbed the population according to certain conditions after each iteration [35]. These methods have shown promising results in improving the performance of the SSA algorithm in solving optimization problems. Compared with the fixed parameters, the dynamic optimization parameters can improve the motion efficiency of the snake robot, and the multi-strategy improved sparrow search algorithm proposed in this paper is superior in optimization efficiency compared with the previous optimization methods.

The main contribution of this paper is to replace the proportional-derivative (PD) controller, which is currently most commonly used to drive and control underwater snake-like robots, with a nonsingular terminal sliding mode controller. In addition, to investigate the tracking of curved routes for underwater snake-like robots under the influence of

unknown currents, we use a multi-strategy improved sparrow search algorithm (MISSA) to speed up convergence and improve global search capability during the tracking process.

To enhance the readability and organization of this paper, the remainder of this manuscript is structured as follows. In Section 2, we present a control-oriented model of a USR and describe the design of the outer-loop controller using the integral light-of-sight (ILOS) method and the inner-loop controller using the non-singular terminal sliding mode control (NSTSMC) approach. In Section 3, we introduce the evaluation criteria for robot movement efficiency, followed by the enhancement of the sparrow search algorithm (SSA) through the inclusion of differential variation, cross-variation, Student distribution, and circle chaotic mapping techniques. We then confirm the effectiveness of the MISSA in terms of search efficiency and precision and use it to dynamically adjust the robot's gait parameters. In Section 4, we present simulations of two paths and compare the optimization effects under several constant parameter conditions. Finally, we conclude and provide recommendations for further research in Section 5.

## 2. Path following Based on ILOS

In this section, we consider a simplified control-oriented model of a USR, as proposed in [36], which utilizes the integral line of sight (ILOS) path following the control method. The controllers include (1) the gait mode controller, which generates a sinusoidal motion pattern to propel the USR forward; (2) the heading controller, which steers the robot in the desired direction angle, enabling it to move towards the desired path and progress along it; and (3) the ILOS guidance law, which produces a desired heading angle, allowing the USR to move towards and along the desired path. The joint coordinates are controlled by an inner-loop nonsingular terminal sliding mode controller, following the creation of reference joint coordinates by the outer loop controller to obtain the desired sinusoidal gait pattern and heading angle $\theta_{ref}$.

### 2.1. Motion Control Approaches of Underwater Snake-like Robot

#### 2.1.1. Notations and Defined Symbols

In this letter, we present a mathematical model of an underwater snake-like robot composed of $N$ rigid links interconnected by $N-1$ active joints. The robot's links are assumed to have an equal length of $2l$ and an evenly distributed mass of $m$, resulting in a moment of inertia $J = 1/3\ mL^2$. The center of mass (CM) of each link is located at the center point, at a length of $l$ from either end. The total mass of the robot is $Nm$. To construct the mathematical model, the following matrices and vectors are utilized in the subsequent sections.

$$A = \begin{bmatrix} 1 & 1 & \cdots & 0 & 0 \\ 0 & 1 & \cdots & 0 & 0 \\ \vdots & \vdots & \ddots & \vdots & \vdots \\ 0 & 0 & \cdots & 1 & 1 \end{bmatrix}, D = \begin{bmatrix} 1 & -1 & \cdots & 0 & 0 \\ 0 & 1 & \cdots & 0 & 0 \\ \vdots & \vdots & \ddots & \vdots & \vdots \\ 0 & 0 & \cdots & 1 & -1 \end{bmatrix}$$

where $A, D \in R^{N \times N-1}$, and matrix $A$ is an addition matrix used to add neighboring elements of a vector, while matrix $D$ is a difference matrix utilized to calculate the subtraction of adjacent elements. Furthermore, $e = [1 \cdots 1]^T$ is a summation vector, used for calculating the summation of all elements of a vector.

Table 1 lists the variables and their definitions that appear in this article.

#### 2.1.2. Complete Control-Oriented Model

The state vector of the USR can be written as follows:

$$x = \begin{bmatrix} \phi^T & \theta & p_x & p_y & v_\phi^T & v_\theta & v_t & v_n \end{bmatrix}^T \in R^{2N+4}$$

where $\Phi \in R^{N-1}$ is a vector consisting of distances between $N$ adjacent connecting links; $\theta \in R$ is the absolute orientation of the USR as a whole; $(px, py)$ is the position of USR's

CM in the inertia frame; $V_\Phi \in R^{N-1}$ are the joint velocities; $V_\theta \in R$ is the angular velocity. $v_t$ is the tangential direction velocity of the USR; and $v_n$ is the normal direction velocity of the USR.

**Table 1.** Symbols and their definition.

| Symbol | Description | Symbol | Description |
|---|---|---|---|
| $\phi$ | Normal displacement between adjacent connecting rods | $N$ | Number of connecting rods |
| $\phi_{ref}$ | Reference normal displacement between adjacent connecting rods | $n$ | Number of path points |
| $\phi_0$ | Joint displacement | $e^T$ | Summation vector |
| $V_\phi = \dot{\phi}$ | Normal velocity between adjacent connecting rods | $\lambda_1, \lambda_2$ | Fluid force coefficient |
| $dV_\phi = \ddot{\phi}$ | Normal acceleration between adjacent connecting rods | $\theta_{cd}$ | The directional angle of sea currents |
| $\theta$ | Robot heading angle | $\theta_{pt}$ | Path tangent angle |
| $\theta_{ref}$ | Robot reference heading angle | $e_s$ | Path tracking error |
| $V_\theta = \dot{\theta}$ | Robot deflection angular speed | $\alpha$ | Serpentine curve amplitude |
| $dV_\theta = \ddot{\theta}$ | Angular acceleration of robot deflection | $\beta$ | Serpentine curve phase shift |
| $u$ | Joint drive force/torque | $\omega$ | Serpentine curve angular frequency |
| $(p_x, p_y)$ | Robot center-of-mass coordinates | $P_i(x_i, y_i)$ | Location of path points |
| $V_t$ | Robot tangential speed | $S$ | Path variables |
| $dV_t$ | Robot tangential acceleration | $S_i$ | Path point variables |
| $V_{t,rel}$ | The relative tangential velocity of the robot | s | Sliding mold surface |
| $V_n$ | Robot normal velocity | $a_0, a_1, a_2, a_3,$ $b_0, b_1, b_2, b_3$ | Polynomial coefficients |
| $dV_n$ | Robot normal acceleration | $\Delta$ | Forward-head distance |
| $V_{n,rel}$ | The relative normal velocity of the robot | $y_{int}$ | The integral role of ILOS |
| $V_c$ | Currents speed | $\sigma$ | The integral gain of ILOS |
| $V_x$ | Current velocity along the x-axis | $k_\theta$ | Joint phase shift rate of change coefficient |
| $V_y$ | Current velocity along the y-axis | $k, \varepsilon$ | Sliding mode convergence law coefficient |
| $c_t$ | Tangential friction coefficient | $\delta$ | Sliding mode surface coefficient |
| $c_n$ | Normal friction coefficient | $p, q$ | Sliding surface coefficient, positive prime, $2q > p > q$ |
| $c_p$ | Propulsion coefficient | $\dot{T}$ | The current speed of the robot |
| $m$ | Single linkage mass | $(x_s, y_s)$ | The position of a point on the design path |
| $M$ | Column vector consisting of the mass of the connecting rod | $(dx_s, dy_s)$ | Virtual particle velocity component |
| $l$ | Half the length of a single connecting rod | $(x_{s^*}, y_{s^*})$ | Virtual particle position |

To establish a more precise mathematical model, the following three assumptions are made:

Assumption 1 assumes that the fluid is viscid, incompressible, and irrotational in the inertia frame.

Assumption 2 assumes that the robot is neutrally buoyant, meaning that its gravity is equivalent to its buoyancy when submerged in water.

Assumption 3 assumes that the current in the global frame is constant and irrotational, with a value of $V_c = \begin{bmatrix} V_x & V_y \end{bmatrix}^T$.

The simplified control-oriented model of the underwater snake-like robot is highly suitable for control design and can be expressed as follows:

$$\dot{\Phi} = v_\Phi \tag{1a}$$

$$\dot{\theta} = v_\theta \tag{1b}$$

$$\dot{p}_x = v_t \cos\theta - v_n \sin\theta \tag{1c}$$

$$\dot{p}_y = v_t \sin\theta + v_n \cos\theta \tag{1d}$$

$$\dot{v}_\Phi = -\frac{c_n}{m}v_\Phi + \frac{c_p}{m}v_{t,\text{tel}}AD^T\Phi + \frac{1}{m}DD^Tu \tag{1e}$$

$$\dot{v}_\theta = -\lambda_1 v_\theta + \frac{\lambda_2}{N-1}v_{t.rel}e^T\Phi \tag{1f}$$

$$\dot{v}_t = -\frac{c_t}{m}v_{t,rel} + \frac{2c_p}{nm}v_{n,rel}e^T\Phi - \frac{c_p}{nm}\Phi^TA\bar{D}\dot{\Phi} \tag{1g}$$

$$\dot{v}_n = -\frac{c_n}{m}v_{n,rel} + \frac{2c_p}{nm}v_{t,rel}e^T\Phi \tag{1h}$$

The actuator forces at each joint of the underwater snake-like robot are denoted by $u \in R^{N-1}$. Further details regarding this model can be found in [36]. As depicted in Figure 1, the locomotion of the robot is modeled using the translational motion of each link, which is typically less complex than the rotational motion. The author of [36] has demonstrated that the control-oriented simplified model exhibits similar quantitative and qualitative behavior to the complex model. Additionally, the overall stability of the robot system has been demonstrated in [37]. In contrast to land-based snake robots, hydrodynamic force is crucial in modeling underwater snake-like robots. To simplify the model, the snake robot is assumed to swim forward at a low enough speed to disregard the additional mass force. This simplified approach has been verified in numerous simulation experiments and yielded feasible results. Therefore, the simulation and energy efficiency optimization of the path following of the snake robot in this paper is based on the simplified model. The simplified model is shown in Figure 1.

### 2.2. Path Planning and ILOS−Based Controller

#### 2.2.1. Control Objective

The goal of the path−following control in this study is to move the robot towards and subsequently along the desired path with a non−zero forward velocity. When following a straight path, the path−tracking error, denoted as $e_s$, is defined as the vertical distance between the center of mass of the USR and the straight path. The coordinate system established in this paper uses the desired straight path as the *X*−axis and the vertical direction of the desired path as the *Y*−axis. Therefore, the straight path−tracking error $e_s$ is represented by the ordinate $p_y$ of the USR in the global coordinate system. In the case of a

curved path, the path–tracking error $e_s$ is the vertical distance between the center of mass of the USR and the tangent at the virtual point. The objective of the path–following control in this paper is to minimize the path–tracking error, approaching zero. Therefore, feedback control laws are formulated to enable the attainment of the control objectives. The structure of the control system is shown in Figure 2.

(1)  Straight path following:

$$\lim_{t \to \infty} p_y = 0 \tag{2}$$

$$\lim_{t \to \infty} \theta = \theta_{cd} \tag{3}$$

(2)  Curve path following

$$\lim_{t \to \infty} e_s = 0 \tag{4}$$

$$\lim_{t \to \infty} \theta = \theta_{pt} + \theta_{cd} \tag{5}$$

where $\theta_{cd}$ is a constant number that is not zero, and $\theta_{pt}$ is the tangential angle of the curve. Both of them are in general non–zero [19], allowing the robot to compensate for the current component.

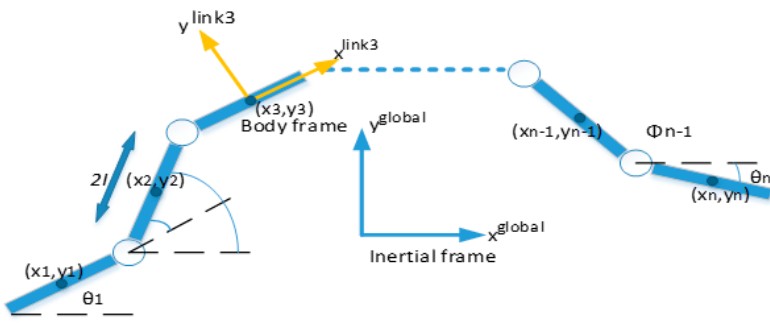

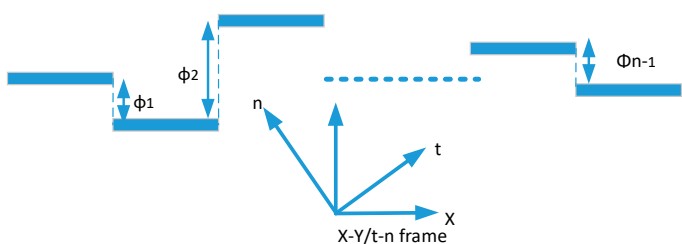

**Figure 1.** Kinematic parameters for both models.

### 2.2.2. Motion Pattern

Similar to biological snakes, snake-like robots typically move forward in a meandering motion, with the shape of the body resembling a sinusoidal curve characterized by regular fluctuations. The connecting link swings due to the reaction force of friction, propelling the robot forward. The serpenoid curve, proposed by Hirose, was the first model to describe the meandering movement of snakes [4]. To achieve the general sinusoidal motion of the USR, each joint of the underwater snake-like robot can track a sinusoidal reference signal, which is one of the most commonly used curve forms. The sinusoidal reference signal is given by:

$$\phi_{\text{ref},i} = \alpha \sin(\omega t + (i - 1)\beta) + \phi_0 \tag{6}$$

In this equation, $\alpha$ represents the maximum displacement of movement between adjacent links, while $\omega$ is the frequency of the sinusoidal motion. The phase shift between adjacent joints is represented by $\beta$, and $\varnothing_0$ is the joint offset that can influence the forward direction of the robot. The joint offset is determined by the direction angle tracked, which is produced by the ILOS law.

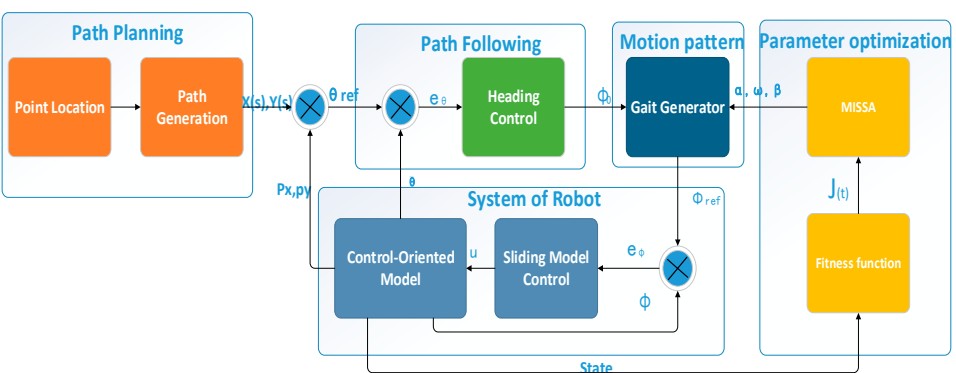

**Figure 2.** The structure of the control system.

2.2.3. Path Planning Based on PCSI

Path planning is a crucial aspect of robot motion planning, particularly for underwater vehicles that must navigate through specific points. One common approach to accomplishing this is using track points to guide the robot to the next point along the given route. For snake-like robots, which are more sensitive to sudden turns in their path, it is essential to plan a smooth path through the waypoints. The piecewise cubic spline interpolation (PCSI) method is particularly well-suited for snake-like robots, as it provides a smooth path with continuous first and second derivatives at the waypoints. This approach is superior to other path-planning methods that lack this level of smoothness and continuity. As such, PCSI is an effective tool for achieving accurate and efficient path planning for underwater snake-like robots.

We can assume that there are $n$ waypoints $P_i(x_i, y_i), i \in \{1, 2, \dots, n\}$ in its path. The path is satisfied with the following conditions:

(1) $S(x_i) = y(i)$ is satisfied on each $P_i$;
(2) There is a continuous second derivative;
(3) There is a cubic polynomial in each $[P_i, P_{i+1}]$

To generate an arbitrary curve, a functional expression for the path is calculated by an introduced parameter $S$, and the coordinates of any point on the path are calculated using the path variable $S$, and the waypoint variable is set to $S_i, i = 1, 2, \dots, n$.

For the generation of spline curves, please refer to [20]. The functions $f(s_i) = (x_s, y_s)$ we obtain are as follows:

$$x_s = a_0 + a_1(S - S_i) + a_2(S - S_i)^2 + a_3(S - S_i)^3 \tag{7}$$

$$y_s = b_0 + b_1(S - S_i) + b_2(S - S_i)^2 + b_3(S - S_i)^3 \tag{8}$$

2.2.4. Outer-Loop Controller

The ILOS approach was first proposed for marine surface vessels in [21]. The integral term in the traditional LOS guidance law is to compensate for the interference of ocean currents. The reference heading angle can be obtained by the ILOS law:

$$\theta_{ref} = -\arctan\left(\frac{e_s + \sigma y_{\text{int}}}{\Delta}\right) \tag{9}$$

$$\dot{y}_{\text{int}} = \frac{\Delta e_s}{(e_s + \sigma y_{\text{int}})^2 + \Delta^2} \tag{10}$$

where the look-ahead distance $\Delta$ is usually set to twice the length of the robot; the $\sigma$ is the integral gain, generally $\sigma > 0$; and the integral action in the guidance law is provided by the $y_{int}$. For more detailed information about the guidance law, please refer to [19]. The schematic diagram of the straight path following is shown in Figure 3a. As shown in [19], the heading controller can steer the heading according to the ILOS angle in

$$\phi_0 = k_\theta(\theta - \theta_{ref}) \tag{11}$$

The constant $k_\theta$ is determined empirically, and $\theta$ represents the directional heading angle of the USR. Although there is no formal proof of stability, numerous simulation results and experimental studies have demonstrated the effectiveness of the controller. The proposed heading controller allows the robot to navigate along the desired path, while mitigating the effects of ocean currents. The path tracing schematic is shown in Figure 3.

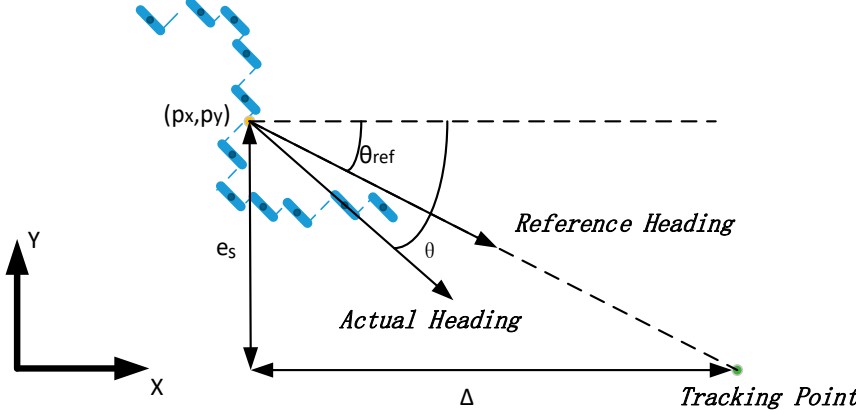

(**a**)

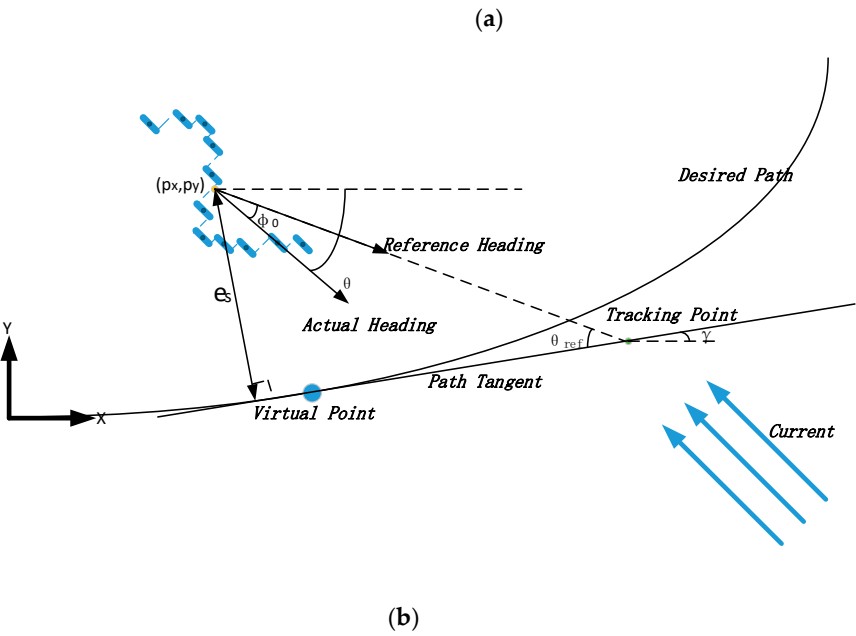

(**b**)

**Figure 3.** (**a**) ILOS guidance schematic diagram for straight path; (**b**) ILOS guidance schematic diagram for curve path.

### 2.2.5. Inner-Loop Controller

The nonsingular terminal sliding mode controller has several advantages, including high resilience to disturbances and unmodeled dynamics, which makes it particularly useful for controlling nonlinear systems. The variable structure control system's straightforward algorithm, quick response time, and ability to resist external noise interference and parameter perturbations make it a popular choice in the field of robot control. The design of the nonsingular terminal sliding mode function is as follows:

$$s = e_s + \frac{1}{\delta} \overset{-p/q}{e}_s \tag{12}$$

where $\delta > 0$ is a design constant; $q < p < 2q$; and $p$ and $q$ are positive odd numbers. The nonlinear term ensures that the system state can quickly approach the equilibrium state in a limited time when it is far from the equilibrium state, and the linear term makes the system state converge quickly when it approaches the equilibrium state. In addition, the synovial function constructed in this way will not have singular points.

In order to let the joint $\varnothing_i$ follow the given reference signal, a nonsingular terminal sliding mode controller is used:

$$\begin{aligned} u = \quad &-M(DD^T)^{-1}(-\frac{c_n}{m}\dot{\phi} + \frac{c_p}{m}V_{t,rel}AD\phi - \ddot{\phi}_{ref} \\ &+ \frac{\delta q}{p}(\dot{\phi} - \dot{\phi}_{ref})^{2-p/q} + ks + \varepsilon\,\text{sgn}(s)) \end{aligned} \tag{13}$$

where $k$ and $\epsilon$ are constants.

The control system designed in this paper is a multi-input, multi-output nonlinear system with joint torque $u$ as input and joint rotation angle/displacement output. The following is a simple analysis of the nonsingular terminal sliding mode controller in this paper.

By choosing the Lyapunov function $V = \frac{1}{2}s^2$, it is obvious that it is positive definite.

$$\begin{aligned} \dot{V} = s\dot{s} \quad &= s(\dot{e}_s + \frac{p}{\beta q}\dot{e}_s^{p/q-1}\ddot{e}_s) \\ &= s(\dot{e}_s + \frac{p}{\beta q}\dot{e}_s^{p/q-1}(-\frac{c_n}{m}\dot{\phi} + \frac{c_p}{m}V_{t,rel}AD\phi + \frac{1}{m}DD^Tu - \ddot{\phi}_{ref})) \\ &= s(\dot{e}_s + \frac{p}{\beta q}\dot{e}_s^{p/q-1}(-\frac{\beta q}{p}(\dot{\phi} - \dot{\phi}_{ref})^{2-p/q} - ks - \varepsilon\,\text{sgn}(s)) \\ &= -\frac{p}{\beta q}\dot{e}_s^{p/q-1}(ks^2 + \varepsilon|s|) \end{aligned} \tag{14}$$

Equation (14) is negative. Therefore, the nonsingular terminal sliding mode controller designed in this paper can be realized, and the system is stable.

### 2.2.6. Improved ILOS Methods for Curved Path Following

To calculate the tracking error of a curved trajectory, a virtual point on the curve is assumed, which is related to the speed of the robot. This virtual point is determined based on the curvature of the curve and the current speed of the robot. That is:

$$\dot{T} = \frac{\sqrt{\dot{p}_x^2 + \dot{p}_y^2}}{\sqrt{\dot{x}_s^2 + \dot{y}_s^2}} \tag{15}$$

where $\dot{p}_x$ and $\dot{p}_y$ are the velocity components of the robot's center of mass along the inertial coordinate axis, respectively. The position of the virtual point is then compared to the actual position of the robot on the curve to calculate the error. The error is defined as the distance between the virtual point and the actual position of the robot on the curve. By continuously calculating the error and making adjustments to the robot's steering, it is possible to minimize the tracking error and ensure that the robot closely follows the curved trajectory.

The normal equation at any point of the path can be obtained from Equations (7) and (8):

$$\dot{y}_s(p_y - y_s) = -\dot{x}_s(p_x - x_s) \tag{16}$$

Assuming the normal equation passes through the CM of the robot and using the Newton–Raphson method, the path coordinates of the virtual point $(x_{s^*}, y_{s^*})$ can be obtained. Hence, tracking errors in the complex path can be written as:

$$e_s = -(p_x - x_{s^*})\sin\gamma + (p_y - y_{s^*})\cos\gamma \tag{17}$$

where $\gamma$ is the tangent angle at the current position of the virtual point $(x_{s^*}, y_{s^*})$. Similar to Equation (11), the reference joint displacement in the curve path can be given by Equation (17).

$$\phi_0 = k_\theta(\theta - \gamma + \arctan(\frac{-e_s}{\Delta})) \tag{18}$$

From the above formula, we can see that the change of $\Delta$ will produce different joint offsets, which will change the degree of robot steering. When USR is far from the path, it is hoped that the value of $\Delta$ will be small, so that the robot can move swiftly toward the desired path. When USR is close to the path, it is hoped that $\Delta$ will be larger, so that the robot can swim along the desired path more stably. Hence, the adaptive $\Delta$ is designed as follows, which is related to the tracking error:

$$\Delta = (\Delta_{\max} - \Delta_{\min})e^{-ke_s^3} + \Delta_{\min} \tag{19}$$

where $k$ is an active constant. Generally, $\Delta_{max}$ = 5$Nl$, $\Delta_{min}$ = 0.8$Nl$, and $l$ is the half length of each link.

## 3. Locomotion Efficiency Optimization Based on MISSA

In this section, we will discuss the efficiency of movement for the underwater snake-like robot, as measured by the ratio of distance traveled to energy consumption. Furthermore, we will explore how this efficiency can be optimized through the use of the MISSA.

### 3.1. Efficiency of Underwater Snake-like Robot

In an underwater snake-like robot, the forward propulsive force is generated by the interaction between the moving connecting link and the surrounding liquid. As a result, the actuator torque input to the joint is converted into a combination of joint motion and energy dissipated by the fluid. The joints are considered to be ideal in this context. Therefore, the energy consumed by the joint rotation is considered to be the total energy consumed by the robot. In the simplified control-oriented model, $u_i$ denotes the moment of the $i$-th joint, while $\varnothing_i$ represents the rotational displacement of the $i$-th joint resulting from $u_i$. As a result, the total energy consumed by the robot within the time $T$ can be calculated as follows:

$$Energy = \int_0^T (\sum_{i=1}^{n-1} u_i(t)\phi_i(t))dt \tag{20}$$

The distance that the USR swims in time $T$ is as follows:

$$Distance = \sum_{t=0}^T \sqrt{(p_x(t+1) - p_x(t))^2 + (p_y(t+1) - p_y(t))^2} \tag{21}$$

Hence, the efficiency can be written as follows:

$$Efficiency = \frac{Distance}{Energy} \tag{22}$$

As demonstrated in Equation (22), given the same amount of energy consumption, a higher efficiency value corresponds to a greater distance traveled by the robot. In other words, a higher efficiency indicates better movement efficiency for the underwater snake-like robot.

It is evident that the distance covered by the robot in a given unit of time is dependent on the speed of the robot during that unit of time. This speed, in turn, is related to the joint motion of the robot, which can be controlled through different combinations of α, ω, and β. The authors of [37] provide detailed information on the relationship between joint swing amplitude and frequency, and the resulting speed of the robot. The energy consumed by the robot can be calculated using Equation (20), which is the product of actuator torque and joint displacement. Moreover, Equation (13) reveals that the value of *u* is dependent on the value of joint displacement, and Equation (6) demonstrates that joint displacement is related to the parameters of α, ω, and β. Therefore, the efficiency of the robot can be determined by its gait parameters, and the optimization of robot efficiency can be achieved by optimizing the gait parameters.

In recent years, the evolution of computer technology has facilitated the development of intelligent algorithms for solving optimization problems, which are constantly updated and refined. This paper addresses the need to dynamically and rapidly identify an appropriate parameter combination for achieving path-following and efficiency optimization under a random curve path. To tackle this problem, we propose an improved sparrow search algorithm based on multiple strategies, which is specifically designed for parameter selection.

## 3.2. Efficiency Optimization with MISSA

### 3.2.1. Principle of the Sparrow Search Algorithm

The classical sparrow search algorithm is a type of bionic optimization algorithm that operates on a heuristic basis. In the context of the parameter optimization problem addressed in this paper, a combination of three parameters is considered as a sparrow. The sparrow changes its position based on a specific update rule and, after a certain number of iterations, arrives at the optimal parameter combination.

The sparrow population can be divided into three distinct groups based on their predation behavior. These groups are producers, scroungers, and watchdogs, which shown in Figure 4.

(a) The producer population constitutes approximately 20% of the total sparrow population. Producers tend to possess high levels of energy reserves that can guide foraging areas or directions for the entire population. In the context of the parameter optimization problem addressed in this paper, the producer corresponds to the parameter combination that exhibits a fitness value that ranks within the top 20% of the population. The update rule for the producer's location is expressed as follows:

$$X_{i,j}^{t+1} = \begin{cases} X_{i,j}^t e^{-i/r*maxgen}, & R < ST \\ X_{i,j}^t + QH, & R \geq ST \end{cases} \tag{23}$$

where *t* indicates the current iteration; the *maxgen* is the maximum number of iterations; *i* is the current number of iterations; *r* and *R* are random numbers; *Q* is a random number that follows a normal distribution; *H* is a $1 \times dim$ matrix, where each element is 1; and *ST* ($ST \in [0.5, 1.0]$) indicates the security threshold.

(b) The scrounger population accounts for approximately 80% of the total sparrow population, and they typically follow producers to locate sources of food. For the parameter optimization problem addressed in this paper, the scrounger corresponds to the parameter combination that does not rank in the top 20% of the population in terms of fitness value. The update rule for the scrounger's location is expressed as follows:

$$X_{i,j}^{t+1} = \begin{cases} Qe^{(X_{worst}^t - X_{i,j}^t)/i^2}, & i > \frac{n}{2} \\ X_{i,j}^{t+1} + \left| X_{i,j}^{t+1} - X_B^{t+1} \right| A^+ H, & otherwise \end{cases} \tag{24}$$

where $X_B^{t+1}$ represents the position of the optimal producer in the $t + 1$ generation; $X_{worst}^t$ represents the position of the global worst individual in the current $t$ generation; and $A^+ = A^T (AA^T)^{-1}$ where $A$ is a $1 \times dim$ matrix, whose every unit at the moment array is randomly assigned $-1$ or 1.

(c) A small subset of the sparrow population, ranging from 10% to 20%, is randomly selected to serve as watchdogs. Watchdogs are responsible for ensuring the diversity and stability of the population. The update rule for the watchdog's location is expressed by Equation (25), as follows:

$$X_{i,j}^{t+1} = \begin{cases} X_{best}^t + \mu \left| X_{i,j}^t - X_{best}^t \right|, & f_i \neq f_g \\ X_{i,j}^t + c \left( \frac{\left| X_{i,j}^t - X_{worst}^t \right|}{(f_i - f_{worst}) + \rho} \right), & f_i = f_g \end{cases} \tag{25}$$

where $X_{best}^t$ is the global optimal value of the current $t$ generation; $\mu$ is the control coefficient of the step size (a normal distributed random number with mean 0 and variance 1); $c$ is a random number from $-1$ to 1; and $f_{worst}$, $f_i$, and $f_g$ are the local worst fitness value, the current local fitness value, and the global optimal fitness value, respectively.

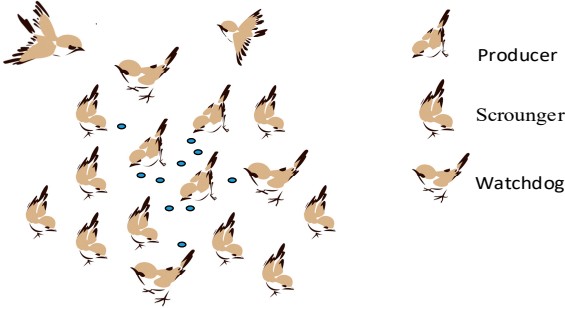

**Figure 4.** Sparrow position schematic diagram.

To simplify the optimization process, the inverse of Equation (22) is used as the fitness function. At each iteration, the improved SSA algorithm returns the best motion parameters for the current period, based on the robot's current state and the current parameter values used for motion. These optimized parameters can then be applied to subsequent movements to enhance the overall performance of the robot.

3.2.2. Multi-Strategy Improved SSA

In this paper, a variety of strategies are applied in the process of sparrow position updating, and the dynamic decrease of shrinkage is realized.

(1) In the initialization phase of the sparrow search algorithm, the random generation method can lead to an uneven distribution of the sparrow population, which can adversely impact the optimization process during later iterations. To address this issue, researchers have proposed the use of chaotic mapping due to its characteristics of randomness, ergodicity, and regularity [38]. Chaotic sequences can be used to initialize the position of individual sparrows. Tang et al. [39] employed the cube chaotic mapping to increase the diversity of the population, and also incorporated the elite reverse learning strategy and sine and cosine optimization algorithm to enhance the sparrow search algorithm, resulting in improved accuracy and stable planning of feasible flight paths for unmanned aerial vehicles (UAVs). However, L. Shan et al. [40], found that tent mapping, although more evenly distributed, has small and unstable

periods that may lead to fixed points. On the other hand, circle mapping is relatively stable and exhibits similar uniformity as the tent map [41]. In this study, we adopt circle chaotic mapping to generate the initial population, which is defined as follows:

$$P_{t+1} = \text{mod}(P_t + 0.2 - \frac{1}{4\pi}\sin(2\pi \times P_t), 1) \tag{26}$$

$$X_{new} = \min_{\text{dim}} + (\max_{\text{dim}} - \min_{\text{dim}})P_{\text{dim}} \tag{27}$$

where *mod* is the remainder function, therefore $P_{t+1} \in (0, 1]$; $min_{dim}$ and $max_{dim}$ represent the minimum and maximum values of *dim* dimensional variables, respectively; and $P_{dim}$ is the value of the *dim* dimension in $P_{t+1}$. The position of the sparrow, denoted by $X_{new}$ is obtained through the circle chaotic map and serves as the initial population distribution for the SSA algorithm. Compared to randomly distributed populations, the improved initial population position distribution is more uniform. This uniform distribution broadens the search range of sparrows in space and increases the variety of swarm positions, which ultimately improves the optimization efficiency of the algorithm. Moreover, this improvement addresses the algorithm's tendency to converge on local extrema. As a result, the algorithm's optimization performance is significantly enhanced.

(2)  To improve the convergence performance of the production population of the algorithm and give the population lower divergence and better stability, this paper carries out crossover and variation on the optimal value of the production population according to a certain probability. Individual producers use binary coding, where the extremum of the individual body crosses the extremum of the population. The crossover probability ($p_c$) is a crucial factor in balancing local and global searches. However, a fixed probability value is commonly used, and the simulation results indicate that an excessively large crossover probability can cause the population to become erratic, leading to a failure to converge. Conversely, an overly low crossover probability can significantly slow the algorithm's convergence and extend computation time. To address this issue, this paper proposes an adaptive adjustment method for the value of $p_c$, as shown in Equation (28):

$$p_c = p_{c,\max} - (p_{c,\max} - p_{c,\min})e^{(f_{\min} - f_{\max})/k_c} \tag{28}$$

where $p_{c,min}$ and $p_{c,max}$ are the smallest and the largest probability of crossing respectively; $f_{max}$ and $f_{min}$ were the individuals with the highest and lowest fitness values, respectively, in the two populations to be hybridized; and $k_c$ is a constant.

To further improve the optimization performance of the algorithm, an individual is randomly selected from the producer group, and the position of the selected individual is transformed into a binary number. A value in the string structure data is then randomly modified with a certain mutation probability (*pm*). The value of pm is as critical to the algorithm as pc. A large pm is not conducive to convergence, while a small pm can result in an insufficient population abundance. To address this issue, this paper proposes an adaptive pm method:

$$p_m = \begin{cases} p_{m,\max} - (p_{m,\max} - p_{m,\min})\frac{f}{f_{\max}}, & f > f_{avg} \\ p_{m,\max} & other \end{cases} \tag{29}$$

where $p_{m,\min}$ and $p_{m,\max}$ are the highest and lowest mutation probabilities, respectively; the population's greatest value of individual fitness is $f_{max}$; and $f$ represents the population's current individual fitness. The proposed method can effectively control the value of pm and adjust it adaptively, enhancing the search capability of the algorithm while maintaining a suitable population size. During the early stages of algorithm iteration, the probability of cross-over and mutation is high, leading to a significant increase in the variety of the producer population. This can be advantageous in exploring new solutions and improving

the algorithm's search capability. However, in the later iterations of the algorithm, reducing the cross-over and mutation probabilities can enhance the discoverer population's capacity for convergence. This strategy allows the algorithm to converge on the optimal solution with a higher degree of precision and efficiency.

(3) In this paper, a Student-distribution perturbation is applied to individuals whose fitness value is less than the population's average fitness value after each algorithm iteration. The probability density of the Student distribution, also known as T-distribution, is defined by a freedom parameter $v$ and is given by the following formula:

$$f(t) = \frac{\Gamma_{(v+1/2)}}{\sqrt{v\pi}\Gamma_{(v/2)}} \left(1 + \frac{x^2}{v}\right)^{-(v+1/2)} \tag{30}$$

where $\Gamma$ is a gamma function. The proposed perturbation method enhances the population's diversity by introducing new individuals with different fitness values. This approach helps to prevent the algorithm from getting trapped in a local optimum and improves its ability to explore the search space more extensively.

At the beginning of the algorithm's development, the Student distribution takes the form of the Cauchy distribution, which is characterized by a smaller peak value at zero compared to the Gaussian distribution. The Cauchy distribution is flatter and slower, making it more suitable for global search. Additionally, the area surrounding the zero point is larger, and as the method progresses, the Student distribution gradually approaches the Gaussian distribution. This feature enables the algorithm to solve problems locally and enhances its ability to perform local searches effectively.

To increase the diversity of individuals with low fitness values in the population and enhance the algorithm's capacity to depart from the local optimal value, a differential evolution algorithm is applied simultaneously for individuals whose fitness value is higher than the population's average fitness value after each iteration. The differential evolution algorithm is a heuristic algorithm proposed by Storm et al. [42]. The differential variation is selected to perturb individuals with better than average fitness, and its mutation strategy is shown in Equation (31).

$$X^{t+1} = X_{best}^t + F(X_1^t - X_2^t) \tag{31}$$

where $X^{t+1}$ is the sparrow position in the $t + 1$ iteration; $X_{best}^t$, $X_1^t$, and $X_2^t$ is the best individual of generation $t$ and two randomly chosen individuals, respectively; and $F$ is the scaling coefficient. The proposed method enhances the population's diversity and improves the algorithm's ability to explore the search space more effectively. The differential evolution algorithm perturbs individuals with higher fitness values and enables the algorithm to search the local optimum in a more targeted manner. The framework of MISSA is shown in Algorithm 1.

---

**Algorithm 1:** The framework of MISSA

---

**Input:**
N: The number of sparrows
N_P: The number of producers
N_S: The number of scroungers
N_W: The number of watchdogs
M_I: The maximum iterations
**Output:**
X_best: The sparrow corresponding to the optimal fitness value
F_best: The optimal fitness value
1: Initialize N sparrows based on a chaotic map and define their relevant parameters;
2: **while** ($t$ < M_I)
3: **for** $i$ = 1: N_P

---

4:    Update producers' locations according to Equation (23);
5: **end for**
6: Follow Steps 3–5 to update the positions of scroungers and watchdogs, respectively, according to Equations (24) and (25);
7: Get the current global optimal and worst location;
8: Calculate the crossover probability $Pc$ and mutation probability $Pm$, respectively, according to Equations (28) and (29);
9: **if** $Pc$ < rand
10:    Convert individual optimal value and population optimal value into binary and cross them;
11: **end if**
12: **if** $Pm$ < rand
13:    Randomly select an individual from the producers, convert it into binary, and change a value in its data sequence;
14: **end if**
15: **for** $i$ = 1: $N/2$
16:    Differential evolution of $N/2$ individuals with high fitness according to Equation (31);
17: **end for**
18: **for** $i$ = $N/2$ + $1: N$
19:    Perform Student-distribution perturbation on $N/2$ individuals with low fitness according to Equation (30);
20: **end for**
21: Get the current new position. If the current position is better than the previous, update it;
22: $t = t + 1$
23: **end while**
24: **return** X_best, F_best

## 4. Simulation Study

This section presents simulation results to investigate the performance of the inner-loop controller and outer-loop controller, as well as the motion efficiency of the USR influenced by ocean currents, as proposed in Sections 2 and 3. A simplified control-oriented model is utilized to evaluate the performance of the proposed controllers. The framework of USR simulation is shown in Algorithm 2. The simulation results provide an in-depth analysis of the effectiveness and robustness of the proposed controllers in ocean currents environments. The results also demonstrate the impact of ocean currents on the USR's motion efficiency, highlighting the importance of implementing an accurate and reliable control system for USRs operating in dynamic marine environments.

---

**Algorithm 2:** The framework of USR simulation

---

**Input:**
$X0$, $Y0$: Horizontal and vertical coordinates of waypoints
$Vx$, $Vy$: The current in the global coordinate system
$N:$ The number of links of USR
**Explanation:**
$S0$: Virtual point values corresponding to waypoints on the planned curve
$S$: Virtual point values of simulation
$u$: Nonsingular terminal sliding mode controller input
Initialize the state of USR and define its relevant parameters.
1: Using coordinates of waypoints $X0$, $Y0$ plan a smooth path according to Equations (7) and (8);
2: **while** ($S(t) < S0(end)$)
3: Using Equation (15), update the value of the virtual point;
4: Using ILOS, calculate reference heading angle;
5: **for** $i$ = 1: $N - 1$
6:    Obtain sinusoidal reference signal with Equation (6);
7: **end for**

---

8: Obtain controller input u with Equation (13);
9: Update USR's status;
10: Update gait parameters of USR with optimization algorithm MISSA;
11: *t* = *t* + 1
12: **end while**

### 4.1. Algorithm Performance Test

In contrast to the prevalent practice of selecting a limited number of common test functions to evaluate algorithm performance, this paper employs both common test functions and the reciprocal of the efficiency of USR as the fitness function to evaluate the effectiveness of MISSA. Additionally, this paper compares the optimization results of MISSA with those of different algorithms. Table 2 presents several common test functions and parameters, and Figure 5 shows the test function diagrams. The main parameters in the algorithm are set as follows: *dim* = 3 (dimension of the function); *Ns* = 30 (number of sparrows); *max_iter* = 100 (maximum number of iterations); *ST* = 0.6 (safety threshold); *Pcmin* = 0.05; *Pcmax* = 0.7; *Pmmin* = 0.001; *pmmax* = 0.7 (maximum and minimum probabilities of crossover and mutation, respectively); to make the simplified model reach a certain accuracy, the range of gait parameters is: $\alpha \in [0.02, 0.13]$, $\omega \in [0.69, 2.61]$, $\beta \in [0.1, 1]$.

**Table 2.** Test function and condition settings for the optimization algorithm.

| Test Function | Expressions | Definition Domain | Optimal Solution | Global Minimum |
|---|---|---|---|---|
| Ackley | $f(X) = -20e^{-0.2\sqrt{\frac{1}{n}\sum_{j=1}^{n} x_j^2}} - e^{\frac{1}{n}\sum_{j=1}^{n}\cos(2\pi x_j)} + 20 + e$ | $[-40, 40]$ | $(0, 0)$ | 0 |
| Griewank | $f(X) = \frac{1}{4000}\sum_{i=1}^{n} x_i^2 + \prod_{i=1}^{n}\cos(\frac{x_i}{\sqrt{i}}) + 1$ | $[-600, 600]$ | (multi-extreme-points) | 0 |
| Rosenbrock | $f(X) = \sum_{i=1}^{n-1}\left[(1 - x_i)^2 + 100(x_{i+1} - x_i^2)^2\right]$ | $[-100, 100]$ | $(1, 1)$ | 0 |

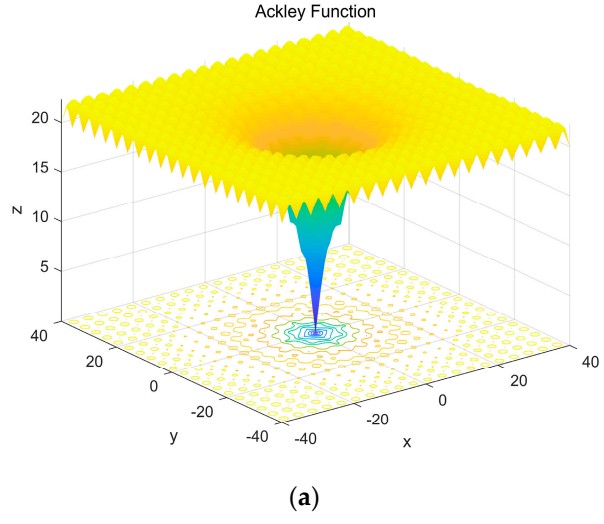

**Figure 5.** *Cont.*

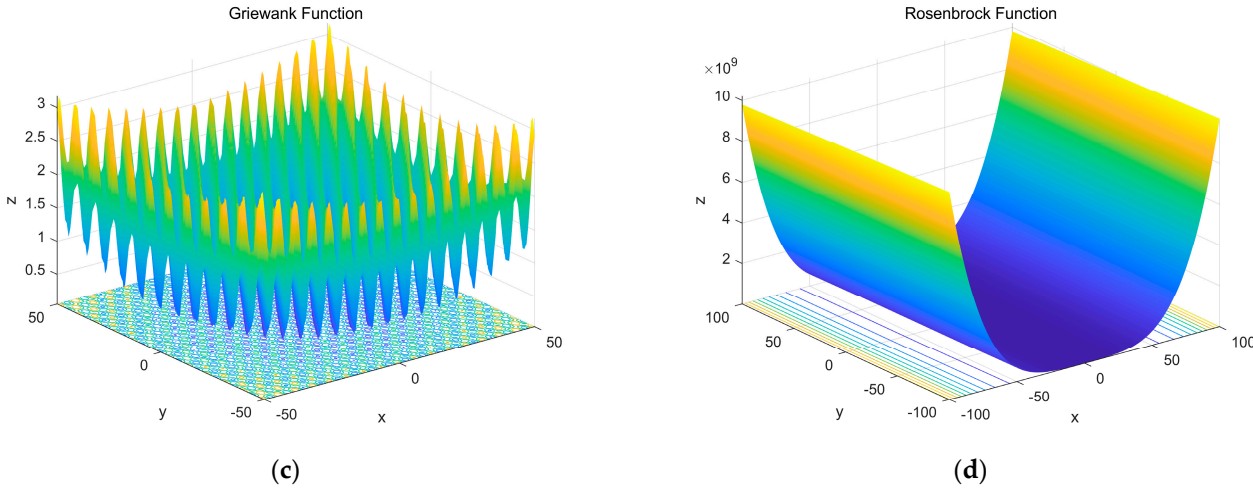

**(c)**　　　　　　　　　　　　　　　　　　**(d)**

**Figure 5.** 3D diagrams of test functions: (**a**) 3D diagram of Ackley function; (**b**) 3D diagram of Griewank function in defined domain; (**c**) local 3D diagram of Griewank function; (**d**) 3D diagram of Rosenbrock function.

In order to measure the tracking level of different algorithms to the expected trajectory, the mean absolute deviation (MAD) is introduced. By taking the absolute value of the errors of each iteration and averaging them, the deviation degree between the actual trajectory and the expected trajectory is compared. The smaller the MAD value, the better the tracking effect. The expression is as follows:

$$MAD = \frac{1}{n}\sum_{i=1}^{n}|x_i - m| \qquad (32)$$

where *n* is the total number of iterations, and m is the expected tracking error, taken as 0.

The sparrow search algorithm [34], improved pigeon-inspired optimization algorithm [33], pigeon-inspired optimization algorithm [43], and particle swarm algorithm [44] are selected for the comparison of the improved algorithms proposed in this paper, and the comparison results are as follows. Table 3 presents the statistical results of the optimization process, including the minimum (Min), maximum (Max), mean, and standard deviation (SD) of the simulation outcomes. The optimization process data has been extracted and analyzed to depict the convergence curve, as illustrated in Figure 6.

**Table 3.** Result statistics of different algorithms.

| Algorithm | Mean | Max | Min | SD | |
|---|---|---|---|---|---|
| Algorithm | 2.556 | 3.366 | 2.540 | 0.013 | |
| SSA | 2.776 | 3.951 | 2.611 | 0.085 | |
| QPIO | 9.923 | 30.558 | 9.462 | 5.193 | Efficiency |
| PIO | $1.16 \times 10^2$ | $1.16 \times 10^2$ | $1.16 \times 10^2$ | $5.05 \times 10^{-27}$ | |
| PSO | 10.399 | 34.550 | 8.909 | 19.574 | |
| Algorithm | Mean | Max | Min | SD | |
| MISSA | 0.076 | 7.599 | $8.88 \times 10^{-16}$ | 0.572 | |
| SSA | 0.198 | 11.125 | $8.88 \times 10^{-16}$ | 1.958 | |
| QPIO | 0.570 | 10.952 | $2.35 \times 10^{-10}$ | 1.650 | F1.Ackley |
| PIO | 0.812 | 9.127 | $4.75 \times 10^{-1}$ | 1.239 | |
| PSO | 1.157 | 13.753 | $2.93 \times 10^{-4}$ | 7.984 | |

**Table 3.** *Cont.*

| Algorithm | Mean | Max | Min | SD | |
|---|---|---|---|---|---|
| MISSA | 1.014 | 1.787 | $8.8 \times 10^{-16}$ | 0.019 | |
| SSA | 1.132 | 3.456 | $8.8 \times 10^{-16}$ | 0.569 | |
| QPIO | 1.011 | 1.372 | $1.4 \times 10^{-4}$ | 0.003 | F2.Griewank |
| PIO | 1.008 | 1.632 | 0.002 | 0.002 | |
| PSO | 1.609 | 21.576 | 0.004 | 8.977 | |
| Algorithm | Mean | Max | Min | SD | |
| MISSA | 8.170 | $4.28 \times 10^2$ | 0.014 | $2.78 \times 10^3$ | |
| SSA | 4.281 | $1.04 \times 10^2$ | 1.752 | $1.88 \times 10^2$ | |
| QPIO | 5.836 | $7.20 \times 10^2$ | 0.155 | $2.72 \times 10^3$ | F3.Rosenbrock |
| PIO | 1.421 | $1.93 \times 10^2$ | 0.283 | $1.85 \times 10^2$ | |
| PSO | 17.030 | $1.30 \times 10^3$ | 0.052 | $1.70 \times 10^4$ | |

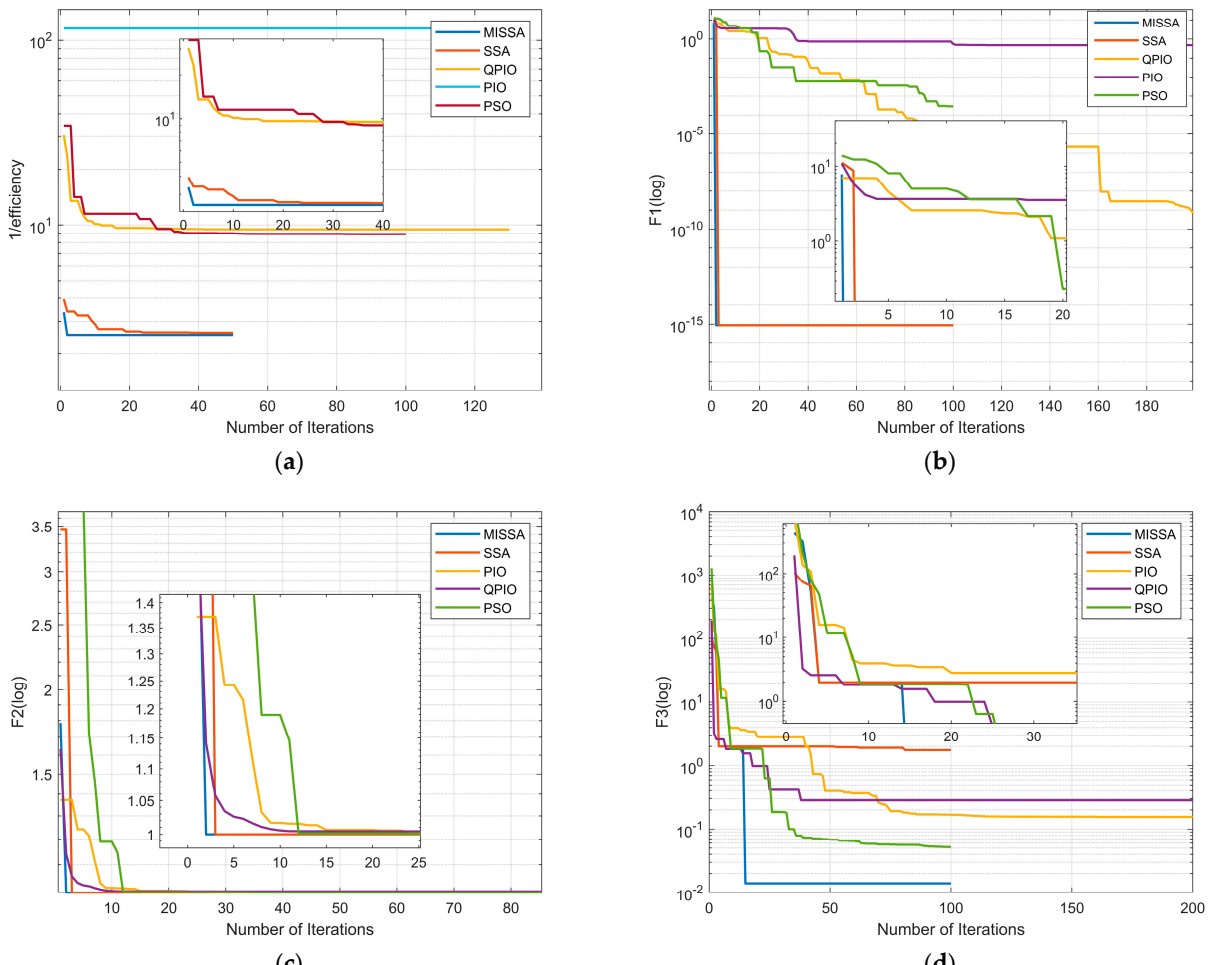

**Figure 6.** (**a**) Optimization performance of five algorithms; (**b**) test results of Equation (1); (**c**) test results of Equation (2); (**d**) test results of Equation (3).

From the data in Figure 6 and Table 2 above, it can be seen that MISSA has the smallest standard deviation (SD) among all algorithms, indicating that MISSA is the most stable. In addition, MISSA converges faster than other algorithms, with smaller orders of magnitude of search results, closer to the global optimal solution of the test function, and has higher

global convergence ability and accuracy. It is worth noting that compared to existing methods that require more iterations, MISSA achieves the same optimization results with fewer iterations. From Figure 6a, it can also be seen that MISSA is more suitable for the snake-shaped robot model in this paper.

### 4.2. Parameter Setting

We consider that the USR consists of $N = 10$ links with the length of each link $2l = 0.14$ m, and mass m = 1 kg. The drag parameters were set as $c_t = 6$, $c_n = 13.5$, $c_p = 23.2$, $\lambda_1 = 6$, $\lambda_2 = 120$. Thus, we can get the distance $\epsilon$ from these values. The control parameters were chosen as $k_\Delta = 1000$, $k_\theta = 0.3$. The irrotational ocean current was assumed as $v_c = [-0.01\ 0.01]^T$ m/s. The initial parameters of gait were chosen as $\alpha = 0.04$, $\beta = 1.57$, $\omega = 0.26$. The parameter for ILOS was set as $\sigma = 0.005$ m/s. The initial state values of the USR were the following: $\overline{p_x} = -1$ m, $\overline{p_y} = 0.4$ m, $\theta = 0$, and $\varphi = 0$. The angular velocity of the USR joints is set to zero.

### 4.3. Path following and Efficiency Optimization

4.3.1. Path following with Adaptive Forward Distance

A great turning curve path is generated by the PCSI method with the waypoints (0, 0), (3, −2), (7, −0.5), (9, 3), (5, 5), and (2, 1), which is shown in Figure 7a with a blue dotted line.

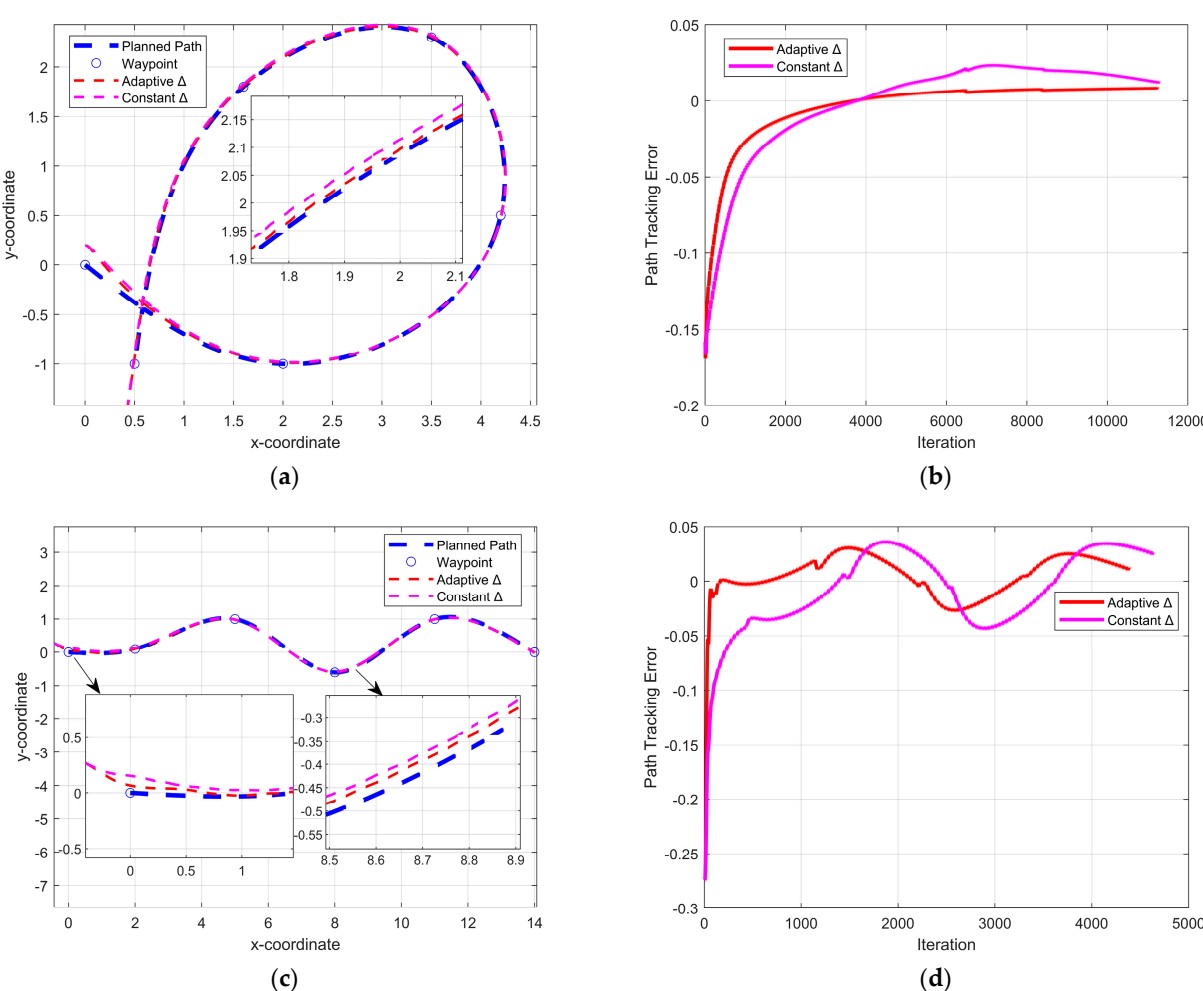

**Figure 7.** Simulation results of different curve paths with the adaptive or constant Δ: (**a**) path following results of USR's center of mass on a large turning curve path; (**b**) path tracking error on a large turning curve path; (**c**) path following the result of USR's center of mass on a continuous turning curve path; (**d**) path−tracking error on continuous turning curve path.

A continuous turning curve path is generated by the PCSI method with the waypoints (0, 0), (2, 0.1), (5, 1), (8, −0.6), (11, 1), and (14, 0), which is shown in Figure 7c with a blue dotted line too.

The effectiveness of the ILOS strategy in enabling the USR to move seamlessly along the intended path is evident in Figure 7a,c. The adaptive forward distance parameter facilitates a quicker approach towards the expected path when the USR is far away from the path, while also enabling better fitting of the expected path when there are sharp turns.

The data in Table 4 reflects the path tracking error corresponding to adaptive $\Delta$ and constant $\Delta$. For the two target curve paths, the MAD of adaptive $\Delta$ is significantly smaller than that of constant $\Delta$, indicating that adaptive $\Delta$ can better track the expected curve path for USR. In addition, as shown in Figure 7b,d, adaptive $\Delta$ can significantly minimize path tracking error and enable USR to stick to the expected path faster while maintaining a smaller tracking error.

**Table 4.** Mean absolute deviations of different curve paths.

| MAD | Large Turning Curve Path | Continuous Turning Curve Path |
|---|---|---|
| Adaptive $\Delta$ | 0.0121 | 0.0144 |
| Constant $\Delta$ | 0.0227 | 0.0282 |

### 4.3.2. Giant Slewing Path

To verify the effectiveness of the control method and optimization algorithm, an expected path with a big turn is obtained by PSCI with the waypoints (0, 0), (3, −2), (7, −0.5), (9, 3), (5, 5) and (2, 1), which is shown in Figure 8a with a blue dotted line.

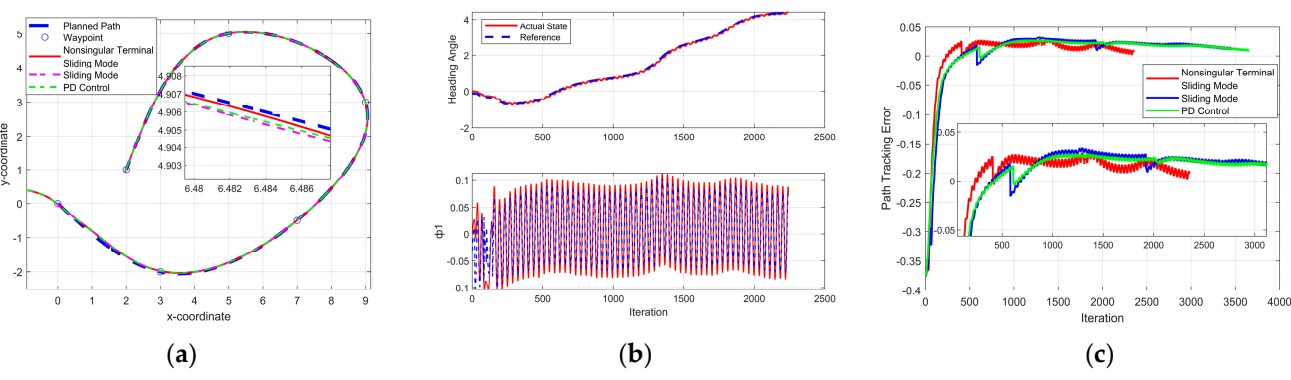

(a)          (b)          (c)

**Figure 8.** Comparison and simulation results of control methods for large turning curve path: (**a**) path following results of USR's center of mass; (**b**) heading angle and $\varphi_1$ of USR; (**c**) path tracking error of three algorithms.

The optimized parameters from the MISSA algorithm were utilized to validate the efficacy of the proposed non-singular terminal sliding mode control approach and were compared to the sliding mode control and PD control strategies, as depicted in Figure 8a. While all three control methods successfully tracked the expected path, the actual path generated by the non-singular terminal sliding mode control method was observed to be more consistent with the expected path at the significant bend. In other words, the proposed method demonstrated superior performance in this task. The heading angle exhibited some fluctuations, while the joint angle of link 1 is displayed in Figure 8b. Notably, the direction angle was observed to lag behind the expected direction angle, which is a normal occurrence. After the expected path turns, the USR adjusts accordingly to reduce the error. The presented figure, Figure 8c, displays the path-tracking errors of the three control algorithms, indicating that the nonsingular terminal sliding mode approach achieves faster convergence to the desired path and lower path-tracking error after stabilization compared to the other two control methods.

We used the MISSA to simulate the motion of USR to verify the effectiveness of the improved MISSA in this paper and introduced four optimization algorithms to compare with MISSA. Table 5 shows the *MAD* of five algorithms. As shown in Figure 9a, compared to other optimization algorithms, the actual path generated by MISSA is closer to the expected path. Figure 9b shows the path-tracking errors of the five algorithms. It can be seen that the path error curve of MISSA has almost no oscillation, which means that the control effect on the actual robot is smoother and more stable, and the error value of MISSA is closer to 0, indicating good tracking performance. Figure 9c,d show the efficiency accumulation and single-time efficiency of the five algorithms, respectively. Among them, the efficiency accumulation of MISSA has always been higher than that of other algorithms. This means that MISSA can find more efficient parameter combinations, and the single-time efficiency of MISSA is relatively stable, unlike other algorithms that occasionally have several high-efficiency outliers.

**Table 5.** Mean absolute deviations of different algorithms for large turning curve path.

|  | **MISSA** | **SSA** | **QPIO** | **PIO** | **PSO** |
|---|---|---|---|---|---|
| MAD | 0.0166 | 0.0173 | 0.0218 | 0.0222 | 0.0171 |

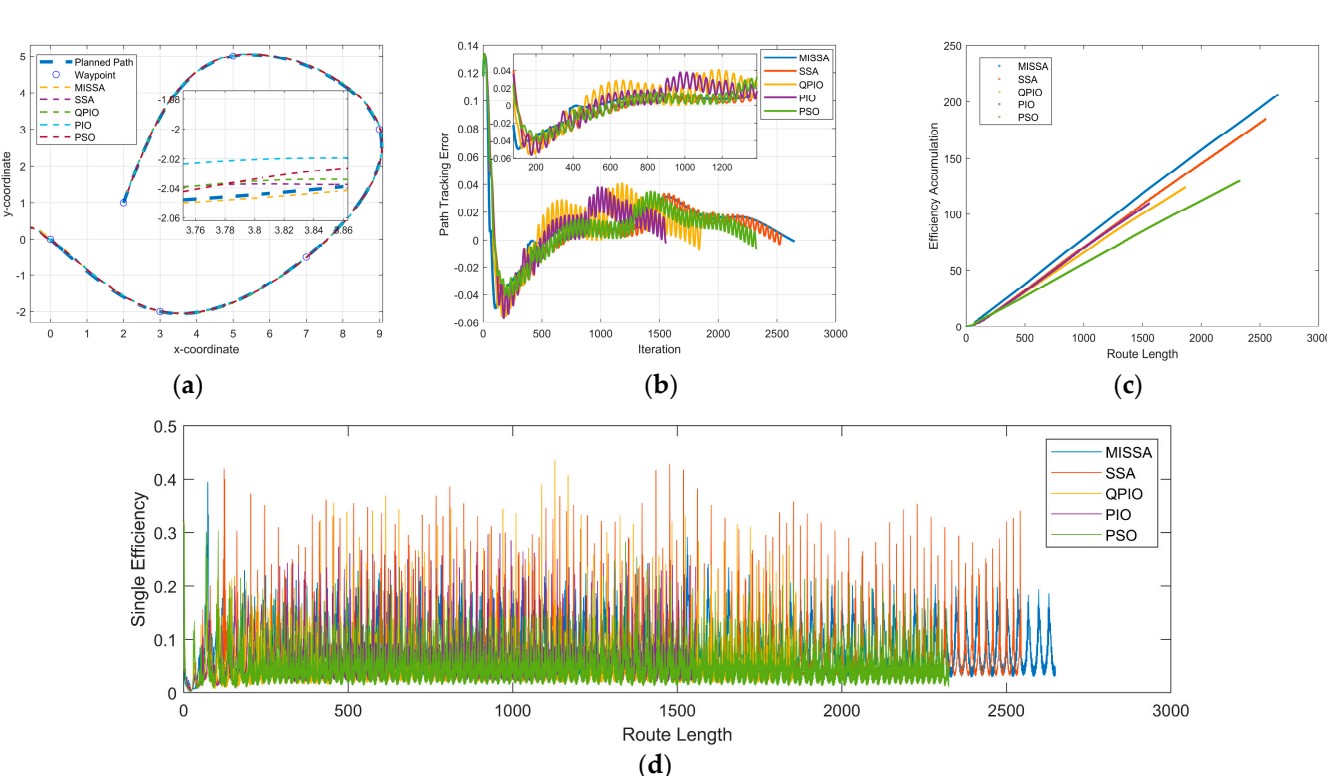

**(a)** **(b)** **(c)**

**(d)**

**Figure 9.** Comparison and simulation results of optimization algorithms for large turning curve path: (**a**) path-following results of USR's center of mass; (**b**) path−tracking error of five algorithms; (**c**) efficiency accumulation of five algorithms; (**d**) single-time efficiency of five algorithms.

There are three constant gait parameter conditions and three optimization parameters used in the large turning curve path in Figure 10a,b. The constant gait parameters are set as (a) Condition1: $\alpha$ = 0.04, $\beta$ = 1.92, $\omega$ = 0.44; (b) Condition2: $\alpha$ = 0.05, $\beta$ = 1.40, $\omega$ = 0.52, (c) Condition3: $\alpha$ = 0.03, $\beta$ = 1.57, $\omega$ = 0.31. To eliminate the influence of algorithmic randomness, Optimization1, Optimization2, and Optimization3 were derived from repeated simulations conducted under identical conditions. As illustrated in Figure 10a, the three optimization methods display comparable trends and achieve higher cumulative efficiency than the other three fixed parameters. Furthermore, Figure 10b demonstrates

that the individual efficiency of the three fixed parameters is lower than that of the three optimization parameters, and the optimized parameters result in a more uniform distribution of efficiency overall.

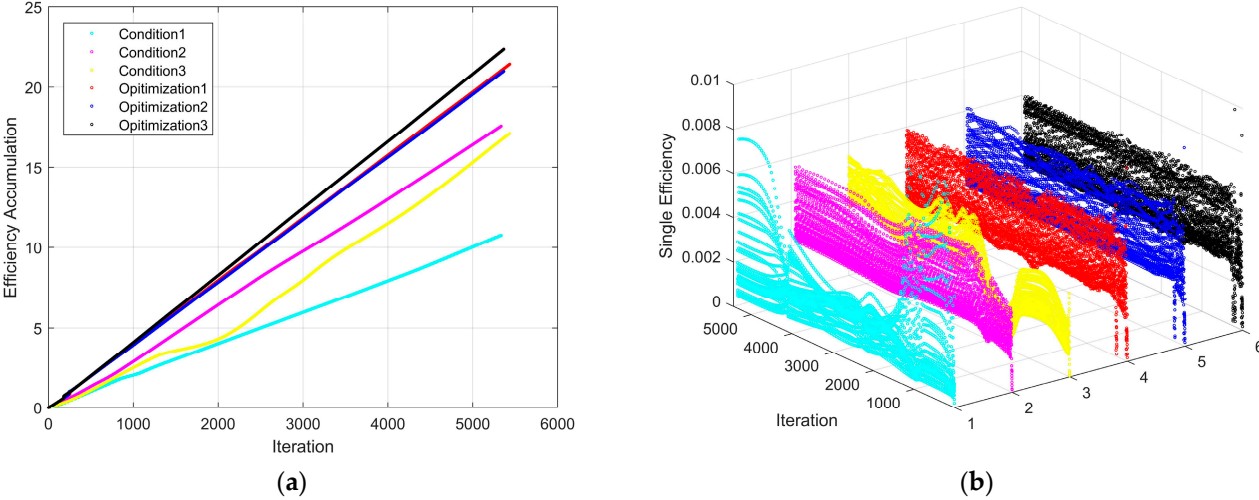

**Figure 10.** Simulation results of large turning curve path: (**a**) efficiency accumulation at different parameters; (**b**) single-time efficiency at different parameters.

### 4.3.3. Continuous Turning Path

To further verify the performance of the proposed control method and optimization strategy, a continuous turning curve path is used as the expected path. The waypoints are set at $(0, 0)$, $(2, 0.1)$, $(5, 1)$, $(8, -0.6)$, $(11, 1)$, and $(14, 0)$, which is shown in Figure 11a. In addition, the start point of the USR's center of mass is $(-0.5, 0.3)$ to verify the effectiveness of the control strategy. Other conditions are the same as before.

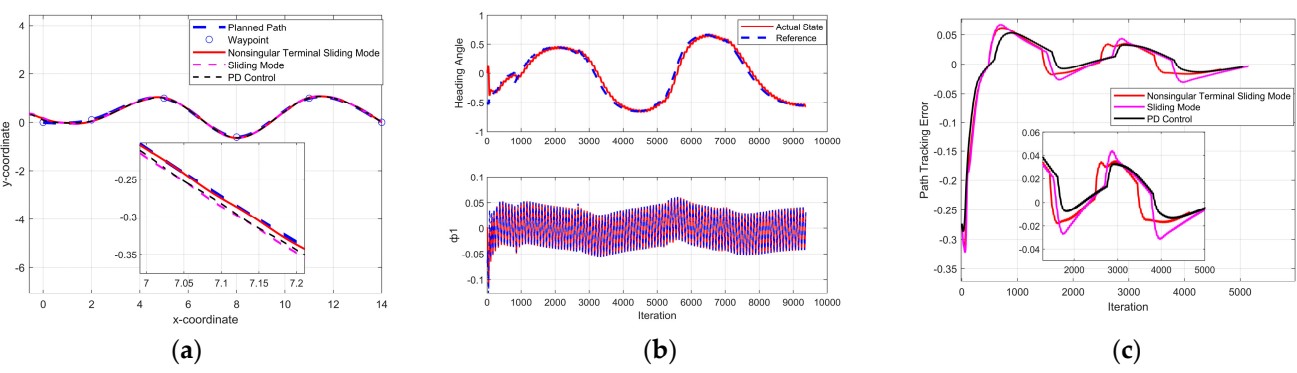

**Figure 11.** Comparison and simulation results of control methods for continuous turning curve path: (**a**) path−following results of USR's center of mass; (**b**) heading angle and $\varphi_1$ of USR; (**c**) path−tracking error of three algorithms.

Upon comparison of the three controllers' path–tracking capabilities in the presence of ocean currents, Figure 11a demonstrates that all three controllers can effectively guide the robot along the intended path. Notably, the non-singular terminal sliding mode controller's USR exhibits superior path-following performance compared to the sliding mode controller and PD controller. Meanwhile, Figure 11b depicts the heading angle and joint angle of link 1, with some fluctuations evident in the heading angle. The USR's heading angle initially undergoes rapid changes from zero to approximately $-0.5$ due to the set initial heading angle of zero, which is rarely observed in actual robot control scenarios. In addition, the direction angle lags behind the expected direction angle, because the path turns before the robot turns. Figure 11c displays the path–acking errors of the three control methods, with

the PD controller and sliding mode controller demonstrating larger fluctuations in tracking error compared to the non−singular terminal sliding mode controller presented in this study. This suggests that the controller proposed in this paper is better suited for handling paths that involve continuous turning.

To verify the effectiveness of the MISSA, we use MISSA and four optimization algorithms to simulate USR tracking a continuous curve path. Table 6 shows the *MAD* of five algorithms. As shown in Figure 12a, compared to other optimization algorithms, the actual path generated by MISSA is closer to the expected path. Figure 12b shows the path–tracking errors of five algorithms. For this continuous curve path, there are significant differences in the error curves tracked by different algorithms. However, it is not difficult to see that the oscillation of the MISSA path error curve is smaller and has a smoother control effect. Moreover, the error amplitude of MISSA is smaller, and closer to 0, and the tracking performance is good. Figure 12c,d show the efficiency accumulation and single-time efficiency of the five algorithms, respectively. Among them, MISSA always has higher efficiency accumulation than other algorithms. In addition, MISSA's single−time efficiency is stable, with higher amplitude than other algorithms.

**Table 6.** Mean absolute deviations of different algorithms for continuous turning curve path.

|  | **MISSA** | **SSA** | **QPIO** | **PIO** | **PSO** |
|---|---|---|---|---|---|
| MAD | 0.0289 | 0.0319 | 0.0472 | 0.0408 | 0.0686 |

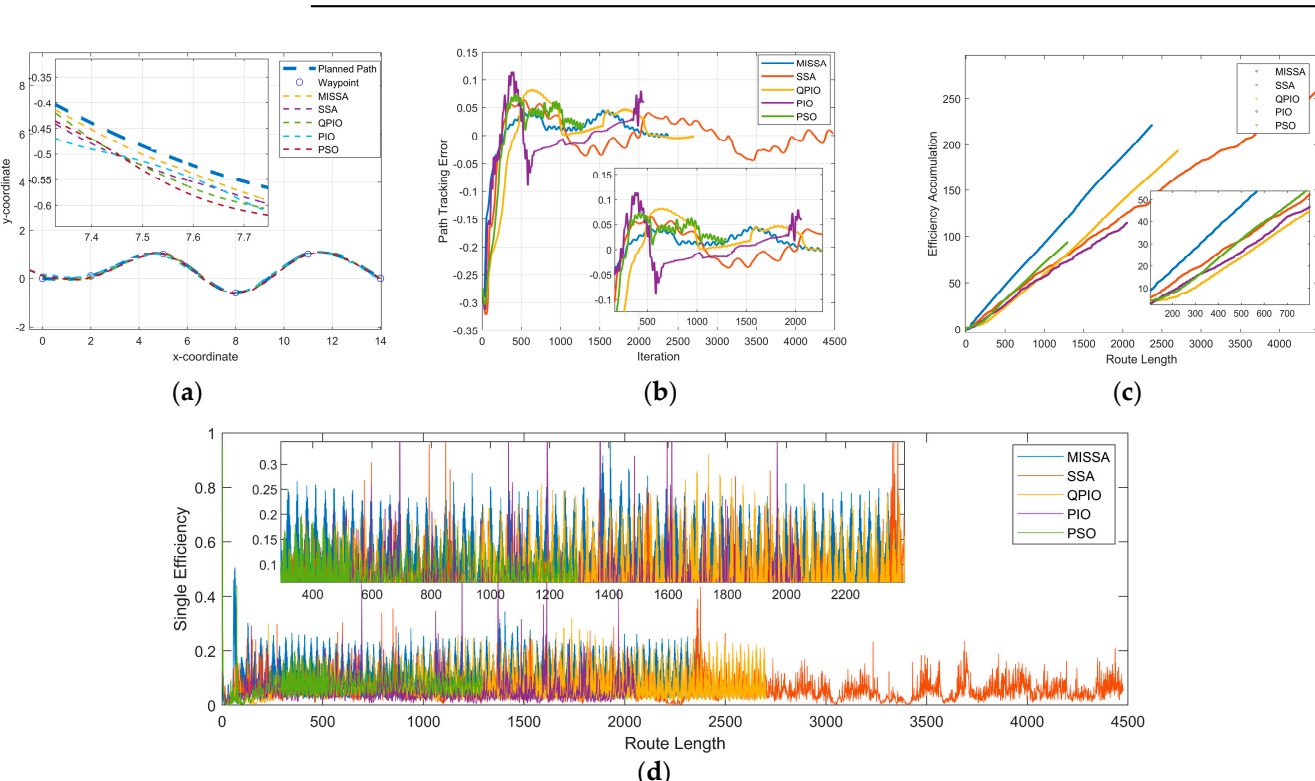

**Figure 12.** Comparison and simulation results of optimization algorithms for continuous turning curve path: (**a**) path-following results of USR's center of mass; (**b**) path-tracking error of five algorithms; (**c**) efficiency accumulation of five algorithms; (**d**) single-time efficiency of five algorithms.

Figure 13a,b display the single efficiency and cumulative efficiency, respectively, of robot movement using three sets of fixed parameters and three MISSA-optimized parameters. The results demonstrate a consistent trend of high efficiency and little difference between single and cumulative efficiency for the three MISSA-optimized parameters, with Optimization2 and Optimization3 displaying sustained high efficiency after a certain period of running time, indicating excellent optimization performance. In summary, all three

MISSA-optimized parameters outperform the three sets of fixed parameters, indicating that MISSA's parameter optimization enhances the USR's movement efficiency.

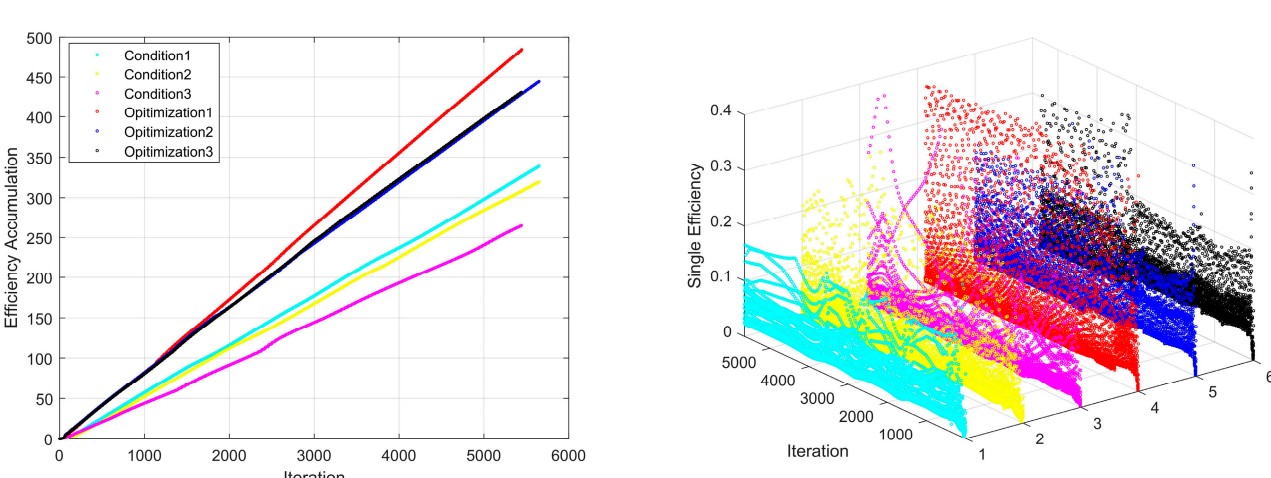

**Figure 13.** Simulation results of continuous turning curve path: (**a**) efficiency accumulation at different parameters; (**b**) single-time efficiency at different parameters.

## 5. Summary

This paper presents a study on the control of curved path following for an underwater snake-like robot under the condition of an unknown constant current. The PCSI method is employed to generate the desired path, with an integral line-of-sight guidance control law utilized in the outer-loop controller for designing the heading controller. Furthermore, a nonsingular terminal sliding mode controller is employed in the inner-loop controller to enable the actual joint angle of the robot to move along the desired joint angle. To maximize the movement efficiency of the robot, an advanced sparrow search method with multiple strategies is proposed to dynamically select gait parameters. The simulation results indicate that the underwater snake-like robot can follow the intended path with minimal deviation, and the multi-strategy enhanced sparrow search algorithm can effectively reduce energy consumption, particularly in situations involving continuous turning or extreme angles.

This paper shows that the paths generated by the PCSI method are suitable for curvilinear path generation for underwater snake-like robots and verifies that the smoothness of the PCSI-generated curves can meet the requirements of non-linear structures such as underwater snake-like robots. In addition, it is demonstrated that the non-singular terminal sliding mode controller can be used to track the curved path of a robot with a complex non-linear system, such as an underwater snake-like robot, even in the presence of currents, and that the control accuracy meets the requirements. It is also demonstrated that the parameters for controlling the joint rotation of a multi-linked robot such as a snake robot cannot be set in stone, but need to be optimized and updated in real time according to the specific conditions.

The current controller proposed in this paper is only applicable in a two-dimensional plane. Moving forward, the main focus of future work will be on developing a method for realizing path tracking in three-dimensional space.

**Author Contributions:** Conceptualization, J.L., H.Z. and H.B.; data curation, J.L. and Y.C.; formal analysis, J.L. and H.B.; funding acquisition, H.Z.; investigation, J.L., Y.C. and H.B.; methodology, J.L. and Y.C.; project administration, J.L. and H.Z.; resources, J.L. and H.B.; software, J.L. and Y.C.; supervision, J.L. and H.Z.; validation, J.L. and Y.C.; writing—original draft, J.L.; writing—review and editing, J.L. and H.Z. All authors have read and agreed to the published version of the manuscript.

**Funding:** This research was supported by the University–Local Integration Category Project "University Vehicles Key Technology R&D Center" and the China Scholarship Council (grant no. 202006680065).

**Institutional Review Board Statement:** Not applicable.

**Informed Consent Statement:** Not applicable.

**Data Availability Statement:** The code for this research algorithm can be obtained by contacting the author by email.

**Conflicts of Interest:** The authors declare no conflict of interest.

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
