# Peer review of "Path Following of an Underwater Snake-like Robot Exposed to Ocean Currents and Locomotion Efficiency Optimization Based on Multi-Strategy Improved Sparrow Search Algorithm"

_jmse, doi:10.3390/jmse11061236_

Round 1
Reviewer 1 Report
Please find out the detailed comments that should be revised:
1. The article is well-written and presented effectively.
2. Based on the optimization methods, the following studies may be useful. Kindly refer:
a. https://doi.org/10.1007/s11042-022-13425-7
b. https://doi.org/10.1142/S0218001422520024
1. The usage of the English language should be rechecked again.
Author Response
Thank you for taking time out of your busy schedule to read our articles carefully, and thank you very much for your recognition of our work. As for the two articles recommended by you, we read them carefully and agree with the contribution made by the author in the two articles. The optimization method of the two articles, which optimized the neural network, has certain help for our article. Therefore, we quoted these two articles in the section of optimization algorithms. The citation is on page 3 of the article Thank you for your recommendation. Wish you all the best!
Reviewer 2 Report
In this theoretical work a control method is designed for the snake robot to perform underwater. It includes the method to create a curve path taking into account currents, and the strategy for tracking how the robot follows the path. The authors introduce a multi-strategies improved sparrow search algorithm to dynamically choose gait parameters that significantly enhance the efficiency of robot movement. The methodology effectively reduces energy consumption by the robot performance, particularly in situations involving continuous turning or extreme angles.
The paper is clearly written, and the figures properly justify the results. I may recommend the manuscript for publication in the Journal of Marine Science and Engineering in the present form.
Author Response
Thank you very much for taking time out of your busy schedule to read our articles carefully. Thank you very much for your recognition of our work. We wish you all the best.
Reviewer 3 Report
This paper provides a multi-strategy algorithm in the form of MISSA for the path-following problem of the robot. In particular, a snake-like underwater robot has been discussed. The quality of the presentation and simulation analysis is good, but there are some aspects of the article, which must be improved as follows:
1) The authors must clearly highlight what are the novel contributions in their proposed study. For example, in bullet points.
2) Figures 6 and 7 are hard to read. Please improve the quality of the figures.
3) Table 2 is good, but would be better to provide the references of the existing techniques and then compare it with your results.
This paper provides a multi-strategy algorithm in the form of MISSA for the path-following problem of the robot. In particular, a snake-like underwater robot has been discussed. The quality of the presentation and simulation analysis is good, but there are some aspects of the article, which must be improved as follows:
1) The authors must clearly highlight what are the novel contributions in their proposed study. For example, in bullet points.
2) Figures 6 and 7 are hard to read. Please improve the quality of the figures.
3) Table 2 is good, but would be better to provide the references of the existing techniques and then compare it with your results.
Author Response
We sincerely thank you for taking time out of your busy schedule to read our article carefully and make some suggestions to make our paper better. Based on your valuable comments, we have done the following:
- As for the problem that we did not emphasize the novel contribution of our article, we rewrote the method adopted by our article at the end of each paragraph of the introduction, and used a separate paragraph at the end of the introduction to emphasize the outstanding contribution made by this article. The revised and added parts are marked in orange in the revised manuscript, while the red part is the deleted part of the previously submitted manuscript. The content Modified is on the pages 2 to 4 of the article.
- As for the picture quality of FIG. 6 and 7 in this article that you proposed is not very clear, we go back to FIG. 6 and 7 and find that there is indeed a little blur. So we did a simulation again, improved the quality of the pictures in Figure 6 and 7, added Table 3, and modified the discourse in the conclusion for your convenience. The content Modified is on pages 18 to 20 of the article.
- Thank you for your recognition of our work. The design of Table 2 is to integrate the results of various algorithms and facilitate readers to compare the data of the article. We have referenced the algorithm mentioned in the table in the paragraph above of the table. In addition to using test functions to test our proposed algorithm as in other articles, the work we do in Table 2 also uses USR's model for testing. The results also show that our improved algorithm is better than other algorithms in the table. The added content is on page 20 of the article.
We would like to thank the referee again for taking the time to review our manuscript.
Reviewer 4 Report
The issues raised by the researchers in the article fall within the thematic area of the Journal of Marine Science and Engineering. However, minor revisions are required before the article is published in the journal. In their work, the authors deal with the implementation of a multi-strategic Sparrow Search Algorithm (MISSA). Meanwhile, References lack an overview of the latest developments in this field. A cursory analysis of publications from the last few years shows that many items are missing. Please fill! The second disadvantage found during the review process was the very complex construction of some sentences. This makes it very difficult to read the text. Please correct the text by a native speaker. The last element is the structure of Chapter 5. Conclusions. This chapter summarises the article's content - the authors do not formulate any conclusions. Therefore, it is necessary:
1) editing the chapter in such a way that it contains the conclusions or
2) renaming the chapter to Summary.
The disadvantage found during the review process was the very complex construction of some sentences. This makes it very difficult to read the text. Please correct the text by a native speaker.
Author Response
We sincerely thank you for taking time out of your busy schedule to read our article carefully and sincerely thank you for your valuable comments.
1.In response to your caution that there are relatively few references to recent years in the original text. We have carefully checked the literature and found that the more classic articles were listed in the initial draft submitted. We have added quotations from articles of recent years to the introduction of the revised manuscript, which has been marked in orange.
2.I am glad to hear your suggestions for writing articles in English. You suggested that the sentence structure was so complex that professionals like you found it difficult to read our article, so we do have some problems with English writing. So we re-read the first draft we submitted earlier. I found that you were right. It is true that what we write is a bit different from what professionals write, but writing and revising is also a way to improve our English writing. We don't want to start by hiring professionals to touch up our English articles, which won't help us improve our English writing. We have tried our best to write this article well in the first draft. We apologize for some of the errors in the article. We have simplified some of the more complex sentence patterns, which we hope will help you read this article. The simplifications have been marked in the article accordingly. Thank you for your understanding and indulgence.
3.In response to your third suggested revision, we found that we did not do a good job in our previous submission of the first draft and did not draw a broader conclusion of applicability from the results in this article. We have therefore added the appropriate content to the revised draft, and the additions are shown below. We will certainly pay more attention to this when writing our next article. We sincerely thank you for these valuable revisions, which have been very rewarding to us.( The location of the added content is on page 24 of the article)
Summary:
This paper presents a study on the control of curved paths following for an underwater snake robot under the condition of an unknown constant current. The PCSI method is employed to generate the desired path, with an integral line-of-sight guidance control law utilized in the outer loop controller for designing the heading controller. Furthermore, a nonsingular terminal sliding mode controller is employed in the inner loop controller to enable the actual joint angle of the robot to move along the desired joint angle. To maximize the movement efficiency of the robot, an advanced sparrow search method with multiple strategies is proposed to dynamically select gait parameters. The simulation results indicate that the underwater snake robot can follow the intended path with minimal deviation, and the multi-strategy enhanced sparrow search algorithm can effectively reduce energy consumption, particularly in situations involving continuous turning or extreme angles.
This paper shows that the paths generated by the PCSI method are suitable for curvilinear path generation for underwater snake robots, and verifies that the smoothness of the PCSI-generated curves can meet the requirements of non-linear structures such as underwater snake robots. Besides, it is demonstrated that the non-singular terminal sliding mode controller can be used to track the curved path of a robot with a complex non-linear system such as an underwater snake robot, even in the presence of currents, and that the control accuracy meets the requirements is also demonstrated that the parameters for controlling the joint rotation of a multi-linked robot such as a snake robot cannot be set in stone, but need to be optimized and updated in real-time according to the specific conditions.
We would like to thank the referee again for taking the time to review our manuscript.
Reviewer 5 Report
Please find my comments and suggestions attached.
Most of the comments are related with the margins, too much space between diagrams, font sizes of the plots and missing spaces.

The document reads well.
Author Response
Thank you for taking time out of your busy schedule to read our article. We are very glad to have professional reviewers like you to read our articles carefully, and we also appreciate your recognition of our articles. Thank you very much for your good suggestions, and in the article for us to diligently mark out our mistakes, let us see the small mistakes we made at a glance. So it's easier for us to modify. Thank you very much. After your reminder, our modification is as follows:
- Blank space is missing before some references, which have been completed.
- Reference [24] is repeated and has been deleted.
- "the underwater robot (USR)" on page 4 has been changed to "USR" and the redundant part has been deleted.
- The "J=1/3ml^2" on page 4 is written by referring to some classical literature, so we did not list it as a separate equation.
- The tables and symbols on page 4 have been modified, and the name of the table has been added above the table.
- The formula on page 6 has been modified according to your suggestion, "(1)" has been modified into (1.1), (1.2), etc.
- The character type and size of the formula on page 10 have been modified accordingly.
- After modification, we put the four pictures in Figure 6 into the same page, and adjusted the font size of Figure 6 and Figure 7. The simulation diagram was redone. And adjust the spacing of the pictures to the appropriate space.
- Also adjust the spacing between the two pictures in Figure 13 to the appropriate spacing.
- The typesetting of the article you sent us is very different from that of the manuscript we submitted. We do not have the problem that the margins you marked on the article are too large. It may be that the typesetting of manuscripts we see is different after systematic conversion. So I submitted to you a pdf version of our manuscript. Please take your time to check if there are any other mistakes.
We would like to thank the referee again for taking the time to review our manuscript.

Reviewer 6 Report

The sentences are too long and ambiguous. in some cases the statements are even contradicting
Author Response
Thank you for taking time out of your busy schedule to read our article carefully and give your valuable comments. In response to your comments or suggestions, we send you a PDF file.

Round 2
Reviewer 5 Report
Dear authors,
Thank you for your quick reply. Please take more time and ensure that you fix the margins.

Author Response
Thank you for taking time out of your busy schedule to read our revised manuscript carefully. Thank you very much for your recognition of our work. According to your comments on our article, we have revised our article accordingly: 1. Formula 15 was re-entered using the formula editor that came with offices software 2. The size and spacing of the two pictures on page 27 have been adjusted. 3. The most important problem is the page margin you mentioned. When we read the article you replied to us, we found that the typesetting of the article is quite different from the manuscript we submitted. This is because the system information on the left side of the first page causes the margins to change. The pictures and tables in this article are centered processing, and the margins of the text should be consistent in theory. But in fact, after the manuscript upload, there is the situation you see. Therefore, we downloaded other articles from this journal for corresponding reading, and found that other articles also had similar problems to our article. Therefore, such problems were caused by systematic typesetting. The margins of some formulas are also adjusted accordingly. Thank you very much for your criticism and correction of our article. We wish you all the best.